



## Jens Esmark's Christiania (Oslo) meteorological observations 1816-1838: The first long term continuous temperature record from the Norwegian capital homogenized and analysed

Geir Hestmark[1] and Øyvind Nordli[2]

1 Centre for Ecological and Evolutionary Synthesis, Department of Biosciences,
Box 1066 Blindern, University of Oslo, N-0316 Oslo, Norway
2 Norwegian Meteorological Institute (MET Norway),
Research and Development Department, Division for Model and Climate Analysis,
P.O. Box 43 Blindern, N-0313 Oslo, Norway

Correspondence to: Geir Hestmark (geir.hestmark@ibv.uio.no)

**Abstract**

In 2010 we rediscovered the complete set of meteorological observation protocols made by professor Jens Esmark (1762-1839) during his years of residence in the Norwegian capital of Oslo (then Christiania). From 1 January 1816 to 25 January 1839 Esmark at his house in Øvre Voldgate in the morning, early afternoon and late evening recorded air temperature with state of the art thermometers. He also noted air pressure, cloud cover, precipitation and wind directions, and experimented with rain gauges and hygrometers. From 1818 to the end of 1838 he twice a month provided weather tables to the official newspaper *Den norske Rigstidende*, and thus acquired a semi-official status as the first Norwegian state meteorologist. This paper evaluates the quality of Esmark's observations, presents new metadata, new homogenization and analysis. The air temperature in Oslo during this period is shown to exhibit a slow rise from 1816 towards 1825, followed by a slighter fall again towards 1838.



## 1 Introduction

The current concern with climate change has increased the interest in early meteorological observation series and evaluation of their quality ( e.g. Bergström & Moberg, 2002; Auer et al., 2007). In a recent paper we analysed the temperature record for the Norwegian capital made 1837-2012 by the astronomical Observatory at the University of Oslo and the Norwegian Meteorological Institute (MET Norway) (Nordli et al., 2015). Previous to 1837 long term observations of the Oslo weather were known to have been made by Jens Esmark (1762-1839), professor of mining sciences at the University of Oslo (then Christiania). A first reanalysis of Esmark's observations was made by meteorologist B. J. Birkeland (Birkeland, 1925). Our rediscovery in 2010 of Esmark's original meteorological observation protocols has provided an opportunity to digitize, homogenize and analyze his data with modern methods.

Esmark is today mostly remembered for his pioneer ascents of many of Norway's highest peaks (Esmark 1802, 1812; Hestmark 2009), his discovery of Ice Ages, and his astronomical explanation of such dramatic climate change as caused by variations in the eccentricity of the orbit of the Earth is now recognized as a precursor of the theories of James Croll and Milutin Milankovich  (Esmark, 1824, 1826; Andersen, 1992; Worsley, 2006; Rudwick, 2008; Berger, 2012; Krüger, 2013). In his own lifetime he was primarily known as a skillful mineralogist and geologist. Throughout his life Esmark maintained a passion for meteorological observation with instruments he crafted himself in accordance with the highest contemporary standards. His main inspiration for this activity were his teachers at Copenhagen University, which he attended 1784-89; first among them the Astronomer Royal, professor Thomas Bugge (1740-1815), who in his observatory tower Rundetårn in the middle of Copenhagen made daily measurements of the weather (Willaume-Jantzen 1896). Esmark also befriended Bugge's instrument maker, the Swede Johan(nes) Ahl (1729-1795) (Esmark, 1825; Anonymous 1839). In addition Esmark followed the lectures of Christian Gottlieb Kratzenstein (1723-1795), professor of medicine and experimental physics, a 'hands on' practical man who enjoyed crafting instruments and all sorts of machines (Snorrason, 1974, Splinter, 2007). From 1789 to 1791 Esmark studied mining sciences at the Norwegian silver town of Kongsberg, and after further studies in Freiberg, Saxony and Schemnitz, Austria-Hungary, he in 1798 moved



back to Kongsberg to take up a position as Assessor in the central mining
administration (*Overbergamtet*) of the dual kingdom Denmark-Norway. At
Kongsberg he also lectured in mineralogy, geology and experimental physics at
the Royal Norwegian Mining Seminar, acting as its temporary Inspector
(Headmaster) from 1799, and permanent Inspector 1802-1815. From 1 January
1799 he three times a day recorded observations of the Kongsberg weather - air
pressure on mercury barometers (in inches and lines), and air temperature in
degrees of Reaumur; documented in a series of small notebooks running
continuously with some lacunae until 16 September 1810, and rediscovered by the
authors in 2010 (Esmark 1799-1810). When Esmark in 1815 moved to the
Norwegian capital Christiania (now Oslo) to become the first professor in the
mining sciences at the University he continued this habit. At least from January
1816 up to and until the day before his death on 26 January 1839 he recorded air
temperature and barometric pressure three times a day. The complete set of his 23
Christiania observation protocols, long believed lost, was rediscovered in 2010 by
the authors, and is now safely deposited in the Norwegian National archives
(Riksarkivet) (Esmark 1816-1838). They provide a unique and detailed picture of
the weather in Oslo in the early 19[th] century. From January 1818 to December
1838 tables of Esmark's observations were published every fortnight in the official
newspaper *Den norske Rigstidende* (cf. Appendix A), and he thus acquired a semi-
official position as Norway's first state meteorologist. Based on a number of
previously unpublished documents (cited as Document 1 etc, with archival
location in Reference list) we here present new metadata for Esmark's
meteorological observations from Christiania, and homogenize, reanalyse and
evaluate his original data with modern statistical tools to characterize the weather
in the Norwegian capital in this period.

**2 Metadata**
**2.1 The location - No. 308, Vestre Rode - Øvre Vollgate 7.**
Esmark's observations were made at his home (cf. Esmark 1823b: *De ere tagne i*
*min Bopel*), and there is no evidence indicating that he changed the location. On 19
August 1815 Esmark was registred as owner of property No. 308 in Vestre Rode
(i.e. Western Quarter), one of the four old quarters of Christiania town (Document



1). It was a modest one-and-half storey house built late in the 18<sup>th</sup> century with an
adjoining a garden. Esmark's continued residence at this address until his death is
documented in annual censuses and tax protocols (Document 2 & 3). Property No.
308 was situated on the north-western side of the street Øvre Vollgate (Øvre
Woldgaden), laid out literally *on* what used to be the outermost western rampart
(*voll*) of nearby Akershus Castle and Fortress (Fig. 1). It was a natural rock
promontory above a meadow to the west where the poor fishing village Pipervigen
would develop later in the 19<sup>th</sup> century, today the site of Oslo Town Hall. In 1815
Øvre Vollgate constituted the south-western limit of Christiania, a town with only
about 15000 citizens (Myhre 1990). Until 1814 the main administration centre of
the dual kingdom was in Copenhagen, but with Christiania in that year acquiring
the new parliament and government after the separation of Norway from Denmark,
the town expanded rapidly. When street numbers were introduced, Esmark's
property was numbered Øvre Vollgt No. 7. The present Øvre Vollgate 7 – an
office highrise – comprises previous numbers Øvre Vollgate 3, 5 and 7.
Esmark's property No. 308 and all neighbouring properties were measured
and mapped for the new matriculation of Christiania in the summer of 1830, and
thus we have very precise data on his house and the surrounding properties at the
relevant time (Document 4). The whole property roughly constituted an elongated
rectangle, approximately 14 m x 60 m (Fig. 2). The unit used in these
measurements was the 'Norwegian alen' (*Norsk alen*), determined by law in 1824
to be 62.75 cm. It was divided into two feet, each divided into 12 inches, each
divided into 12 lines. No. 308 was measured to 2026 square alen, of which the
house (including a yard) was 733 ½ and the garden 1292 ½ square alen (1 square
alen = 0.3937 m$^2$). Thus the whole property was ca. 800 m$^2$, and the house
(including yard) ca. 290 m$^2$.  The house had a 22 alen 6 inch (ca. 14 m) long
façade towards the street Øvre Voldgate, constituting the south eastern border of
the property, with windows, doors, and a gate leading in to the back yard (Fig. 3).
Øvre Vollgate street runs from SW to NE at an angle of roughly 32$^{\circ}$ NE (400
degrees). At the back the house surrounded a small yard, with a narrow passage
opening out to the garden in the NW. As it would have been hazardous to place the
meteorological instruments on the street-side of the house, where passers-by could
have tinkered with them, it is almost certain that they were placed in Esmark's
back yard, a well guarded space.  When the house was finally demolished in 1938,



it was in such bad condition that the Oslo city health authorities demanded the
whole property be sprayed with hydrocyanic acid and that none of the fungus-
infected material be used for construction elsewhere (Document 5).

Esmark's garden on the NW side of the house and court yard was a

continuous slope, dropping ten alen (6,25 m) down along 66 alen length towards
Pipervika. Here it was most probably limited by a fence towards the Præste Gade
street which later changed name to todays Rosenkrantz gate. In 1841, a couple of
years after Esmark's death, most of this garden was indeed sectioned out and sold
to form the new property Rosenkrantz gate 26. In Esmark's time, however, the
promontory remained an open garden space. His neighbours on both sides (No.
307 and No. 309) had the same arrangement of house and garden, with facades to
Øvre Vollgate and gardens sloping down on the back to Præstegaden (Document
6). To the north of the lowermost part of Esmark's property was an open space
called Jomfru Wold's Løkke (No. 368). South of this lower part of the garden was
the street Pipervigbakken, leading down from Rådhusgaten street passing by the
outer ramparts of Akershus fortress and Castle. The sea with Pipervika bay
(Piperviks Bugten) was less than 200 m south of Esmark's garden. His garden was
not an entirely constant environment. In 1823 for instance, he received several
fruit trees from a Danish friend which he planted in the garden (Document 7).

It was a modest residence for a professor, situated in a comparatively poor

part of town, with mainly craftsmen, tradesmen and artisans in the neighbourhood
(Myhre 1990: 40). Here Esmark, a widower since 1811, moved in with his three
sons Hans Morten, Petter and Lauritz, a maid and a manservant (Document 2 & 3).
His daughter Elise resided with her grandparents in Copenhagen, but later returned
to Norway to take up residence in No. 308.

**2.2 The observers**
The great majority of the Christiania observations were made and noted down by
Esmark himself who has an easily recognizable handwriting. His position as
professor in the mining sciences did however sometimes cause him to leave town
on short or long field excursions, some lasting several months. He was away from
Christiania on long voyages in 1818 (Hallingdal), 1819 (Kristiansand); 1822
(Bergen), 1823 (round-trip south Norway), 1826 (Setesdalen), 1827 (Trondhjem)
and 1829 (Copenhagen). In his absense his sons seem to have been instructed to





continue daily observations, and there are extremely few missing data points. The
oldest son Hans Morten Thrane Esmark (b. 1801) in 1825 became a chaplain in
Brevig and moved from Christiania; Axel Petter (b. 1804) became a sailor and was
often away from home; Lauritz Martin (b. 1806), later a professor of zoology at
the Christiania university, and daughter Elise Cathrine (b. 1800) remained at home
until Esmark's death. The sons evidently did not fully share their father's passion,
and although instrument readings were meticulously maintained, the qualitative
notes on weather are often restricted to a single word in Esmark's absence. A
claim (Birkeland 1925: 5) that the botanist Martin Flor performed the observations
in Esmark's absence has not been substantiated, and anyway Flor committed
suicide in 1820.

**2.3 The hours of day**
Esmark's Christiania observation protocols do not indicate the precise hours when
the observations were made. The columns are given as morning, noon and evening
(*Morgen, Middag, Aften)*. A note on the first published table in *Den norske*
*Rigstidende* on 24 January 1818, also says *Morgen, Middag og Aften* without
further specification (Fig. 6). In a summary table of 15 years (1818-1832)
published 1833 Esmark is more explicit: "The barometer observations have been
made daily in the morning, afternoon and evening; in later years at 8 ½ o'clock
morning, at 3 ½ o'clock afternoon and 9 ½ o'clock evening; thermometer
observations at the same times in the afternoon and evening and in the morning
with the help of the night thermometer. From this the middle hight is taken."
**(***Barometerobservationerne ere dagligen gjorte om Morgenen, Eftermiddagen og*
*Aftenen; i de senere Aar Kl. 8 ½ Morgen, Kl. 3 ½ Eftermiddag og Kl. 9 ½ Aften;*
*Thermometerobservationerne paa samme Tider om Eftermiddagen og Aftenen og*
*om Morgenen ved Hjælp af Natthermometret. Heraf er taget Middelhøiden*.)
(Esmark 1833: 235). Thus 8.30 AM, 15.30 (PM), 21.30 (PM). The hour 3 ½ PM
probably coincided with Esmark's return to his house from the lectures at the
University just a few blocks away. The phrasing "in later years" suggests that the
hours had not been constant throughout the whole series. This problem we analyse
further below. Also that a night-thermometer (for measuring minima) was
introduced some time after the start of the series.



**2.4 The instruments and their position**
In a note to his first table presented in the journal *Den norske Rigstidende*, on 24
January 1818, Esmark provides a few details of his measurements: "The
observations are made 34 Rhinelandic feet [i.e. 10.68 m] above the sea, and are the
middle value of observations made morning, noon and evening. The barometer
heights are corrected as they would have been if the barometer was subject to a
temperature of $0^o$. The thermometer hangs freely against north." *(Observationerne*
*ere anstillede 34 Rhinlandske Fod over Havet, og ere Middeltallet af*
*Observationer, anstillede Morgen, Middag og Aften. Barometerhøiderne ere*
*corrigerede saaledes, som de skulle være, dersom Barometret havde været udsat*
*for $0^o$ Temperatur. Thermometret hænger frit imod Nord.*) (Fig. 6).  Esmark also
notes for these (average?) data that "The barometer height is reduced to $0^o$ R. If
one wants it reduced to sea level, one must add a line or 1/12 of an inch to its
height, so that the barometer height at sea level becomes 28.1,20 in French
measure." (*Barometerhøiden er reduceret til $0^o$ R. Vil Man have den reduceret til*
*Havets Overflade, maa Man til den anførte Høide lægge en Linie eller 1/12 Deel*
*af en Tomme, saa at Barometerhøiden ved Havets Overflade bliver 28.1,20 i*
*Fransk Maal.*) (Esmark 1833: 235).

*Thermometers*. Esmark all his life used the Reaumur scale; "R".  The precision of
his Reaumur thermometer was 1/2 of a degree. On a table of averages for the years
1816-1822 Esmark notes: "The thermometer observations are made in shadow in
free air with a Reaumur thermometer, which boiling point is determined at 28
inches 2 lines (French measure) barometric height."
(*"Thermometerobservationerne ere gjorte i Skyggen i fri Luft med et Reaumurs*
*Thermometer, hvis Kogepunkt er bestemt ved 28 Tommers 2 Liniers (fransk Maal)*
*Barometerhöide."*)  (Esmark 1823).

*Barometer*.  Of the barometer used Esmark (1833: 235) states: "The barometer is a
simple barometer, the tube of which is 2 ½ line in diameter and which capsul is 40
lines in diameter, and calibrated after a hevertbarometer." (*Barometret er et*
*simpelt Barometer, hvis Rør er 2 ½ Linie i Diameter og hvis Capsel er 40 Linier i*
*Diameter, samt justeret efter et Hævertbarometer.*)



**2.5 The protocols and data recorded**

Esmark's Christiania protocols are handmade, folded sheets of white paper cut up
and sewn in with a thin grey cardboard cover, one protocol for each year (Fig. 4),
23 protocols in all (Esmark 1816-1838). Esmark interfoliated the official printed
*Almanach* for Christiania. This had for each month 16 days on each page, and thus
Esmark wrote down his data for 15 or 16 days on the first page of a month and the
remaining days from 17 to 28, 29, 30 or 31 on the next page (Fig. 5). The protocols
start on 1 January 1816 and end 31 December 1838, only 26 days before his death;
altogether 8401 days of continuous measurements. There are only a few small
lacunae. Photographs of all the protocols are available at MET Norway (Klimadata
samba server, HistKlim skanna dokument), and digitized values might be
downloaded from MET Norway's home page: http://www.met.no. Esmark
continued observations in January 1839 until the day before his death 26 January,
but these observations are only known through the newspaper *Morgenbladet*,
which had published Esmark's daily measurements since 1834.

Three times a day Esmark recorded temperature to a half degree, and air

pressure with one or two decimals (Fig. 5). In the right hand margin he noted the
weather (*Veirliget*) with qualitative terms; see also Esmark (1833). He used a fairly
limited number of categories: *Precipitation*: *lidt Regn (a little rain)*; Fiin Regn
(drissle); *Regn* (rain); *Regn Bygger/Bÿgger* (showers); *Regn af og til* (Rain now
and then); *megen Regn* (much rain); *Sne* (snow); *Sne Flokker* (snow); *Sne Bygger*
(snow showers). *Cloud cover*: *Klart* (clear), *enkelte Skyer* (a few clouds); *tynde*
*Skyer* (thin clouds); *skyet* (cludy); *skyer i Horizonten* (clouds in the horizon); *disig*
(haze); *Taage* (fog). The most common category was *tykt* (thick) which means a
grey day with haze, often with precipitation.  *Wind*: Wind direction was usually
recorded only once a day, at midday, with categories N, S, V and O, and
combinations , e.g. N. O. (nord ost/north easterly). *Other*: *Torden* (thunder);
*Nordlys* (northern lights); *Flekker i Solen* (sunspots); one or two circles around the
sun; *Høyt vand* (high sea level). In June 1818 Esmark introduced a new parameter:
*precipitation*, measured with a rain gauge, and in the June summary, he could
announce: "In this month there has, according to the rain gauge, fallen rain to a
height, which, if it had been standing, had constituted a height of 1 inch and 9 and
7/12 line. The rain gauge is situated 15 feet above sea level." The low altitude of
the rain gauge suggests that it was placed at the lower part of the slope in his



garden. In October 1820 he presented the readers of *Rigstidende* to his new design
for a hygrometer – an instrument to measure air humidity (Esmark, 1820). It was
modified from a model developed by John Livingstone, and M.D. from Canton,
China, published in the *Edinburgh Philosophical Journal* in 1819 (Livingstone
1819). The general idea was to put a moisture absorbing/releasing chemical
substance (Livingstone used pure sulphuric acid, which was also used to produce
ice) on one side of a balance, balanced against a weight on the other side. The
balance was placed under a glass jar open in the bottom to let air freely flow in and
out, and to protect it from precipitation. Esmark made two new hygrometers
according to this model. "Anyone who desires to see these hygrometers, can see
them at my house" *("Enhver, som har Lyst dertil, kan see disse Hygrometere hos*
*mig.")*(Esmark, 1820)  He had tested them for several months, and thought they
could be used by farmers to predict weather change as substitute for barometers.
He did not, however, use the hygrometer data for his meteorological tables. For the
year 1821 he presented more regular monthly data on precipitation in inches –
from 1 May through October – apparently the months without frost.

**2.6 The published tables**
Starting on Saturday 24 January 1818, with a table presenting weather data for the
first half of the month, the semi-official daily *Den norske Rigstidende* published
Esmark's meteorological observations, which thus acquired an official air. (Fig. 6).
It became a regular series, published twice a month – one table for the first half of
the month, one for the second half – a total of 24 tables each year, all with the
same title "Meteorologiske Iagttagelser i Christiania [year], anstillede af Prof.
Esmark." (Meteorological observations in Christiania [year], made by Prof.
Esmark) etc.. This series running from 1 January 1818 to 15 December 1838 is
absent from all previously published bibliographies of Esmark's works, but in fact
runs to no less than 503 published tables (!) (Appendix A). They present 7665 days
of continuous observations. In addition comes the two full years of 1816 and 1817,
only published summarily by Esmark (1823) but with complete record preserved
in the original protocols. The whole year 1818 was summed up on 8 January 1819
with means etc., and here Esmark also compared the Christiania data to those
obtained by Wargentin in Stockholm, by Bugge in Copenhagen, and (no
observator given) in St. Petersburg, Russia.  It was not a weather forecast but



rather a weather 'backlog', and this may have dimmed their public interest
somewhat. The data given in these published tables differ from the raw data of the
protocols by being daily averages. For each day he gave the barometric pressure
and temperature, averaged from observations made in the morning, at noon, and in
the evening (at first without further precision of hour). To calculate these averages
he apparently used the formula:
$T_m = {}^{1}\!/_{4}\,(T_I + 2T_{II} + T_{III})$                                   (1)
where $T_m$ is Esmark's daily "mean" temperature, and $T_I$, $T_{II}$, and $T_{III}$ are the
observed temperature morning, noon and evening, respectively. To the tables for
the second half of each month, he also appended a note with the mean barometric
pressure and temperature for the entire month, and indicated which days had the
maximum and minimum air pressure and temperature. The mean temperature was
given to 1/100[th] degree (a spurious precision). The series continued in 1820, now
also with the daily wind direction.  Esmark evidently trusted only himself to
calculate the averages and set up the tables, and thus the readers of *Rigstidende*
sometimes had to wait for months to read the weather for the last fortnight. From
1834 Esmark's observations were also published in the Christiania newspaper
*Morgenbladet* every day, with two days delay, i.e. observations for the 1[st] day of
the month were published on the 3[rd] etc. This was initiated after Christiania
doctors suspected a connection between the weather and the cholera epidemics
which struck Norway from 1833 and forward.

**3 Methods**

**3.1 Homogeneity testing**
A homogenous climatic time series shows variations in climate without being disturbed by
other factors involved, like changes in the environment, observational procedures or
instrument calibration. For the study of climate variations the use of homogenous series is of
paramount importance, otherwise the climate analysis might be wrong (e.g. Auer et al., 2007;
Moberg and Alexandersson, 1997; Tuomenvirta, 2001). For testing the homogeneity of
Esmark's temperature series we selected the Standard Normal Homogeneity Test (SNHT) that
has been widely used for testing of both precipitation series and temperature series
(Alexandersson, 1986; Alexandersson and Moberg, 1997; Ducré-Robitaille et al., 2003). The





first version of the test (Alexandersson, 1986) had one step change as the only possibility,
whereas in the version of 1997 both double shifts and a trend were possible outcomes of the
test. In any year the significance of a potential break is examined. The testing followed the
principle of comparing a candidate series (the series under testing) against a reference series.
The reference might be series from one or more neighbouring stations. A candidate series
might also be observations at one particular time of the day, which are compared with other
observation times for the same station. In the latter case we call it "internal testing". Without
contemporary neighbouring stations internal testing is the only possibility. If no significant
break occurs the series is considered homogenous. Esmark's station at Øvre Vollgate 7 as
well as other observation stations used in this article are given in Table 1, with their national
station number (identifier) and name. Before the analysis started all observations were
calculated from degree of Reaumur to degree of Celsius by multiplying Esmark's Reaumur
readings by the factor 1.25.

**4 Results**

**4.1 Homogeneity testing**
For much of Esmark's period of observation there was no other nearby station in operation so
internal testing was the only possibility. The testing was performed both for seasonal (see
Table 2) and monthly (see Table 3) resolutions where observations taken in the morning (I),
noon (II) and evening (III) were compared with each other. By comparing several test results
it was possible to decide at which observation time a shift (inhomogeneity) occurred. Most
striking are the huge shifts detected in spring, summer and autumn when the morning
observation was involved. The most probable year for the shift was 1827; in particular this
was true for the single shift test. Here we apply the common convention to define the shift
year as the last year before the shift. We have to conclude that the morning observation is
inhomogeneous. A further investigation of the daily observations (not shown) suggested that
the change took place within the month of March 1828.

When evening observation was tested against the midday observation a shift seemed to

occur in 1820 or 1821, most probably in 1821. But this break in homogeneity was much less
than that of the morning observation. The shift seems to be absent or very weak during winter
so exact dating was impossible. For convenience the end of 1821 was adopted as the year of
the inhomogeneity.



Tests including the midday observation revealed no additional shifts than those
already detected. The occurrence of the shifts in the tests I vs II and III vs II seemed to reflect
shifts either in the morning or in the evening observations. For the winter season a shift in the
last part of the series was detected, possible shift years were 1832, 1833 or 1834.
The large shift in the morning observation could have masked possible smaller shifts in the
series on both sides of this shift. Therefore the single shift SNHT was applied on two different
parts of Esmark's series: 1816.01-1828.02 and 1828.03-1838.12, parts 2 and 3 in Table 2.
However, no further shifts in the series were detected. The shifts detected in part 1 in the
evening observations of 1821 and in the morning observation in the 1830s for the winter
season were confirmed.
The reliability of the results was further tested on monthly resolution and also
evaluated by comparison with the metadata. Esmark (1833) tells that he uses "a night
thermometer" for the morning observation. Our hypothesis is that in Esmark's terminology
"night thermometer" means "minimum thermometer", and that the introduction of the
minimum thermometer is the reason for the shift in March 1828. This hypothesis was tested
by studying the difference between Esmark's evening observation and the morning
observation the following day for the three homogenous intervals (see Table 4) (the winter
inhomogeneity in the 1830s was ignored). For comparison this was also done for the
observations at the modern station Oslo – Blindern. In the earliest interval (row 1) the
differences in Esmark's observations were very much smaller than those from Blindern, so it
is impossible that Esmark could have noted the nightly minimum temperature in the column
for the morning observation. In the next interval (row 2) the differences are somewhat larger,
but far too small compared to Blindern so the same conclusion has to be drawn: no minimum
thermometer was in use. However, in the third interval (row 3) the differences are nearly the
same as those for Blindern. Even the monthly variations throughout the year are realistic. We
conclude that Esmark for the morning observation used a minimum thermometer in the period
1828.03-1838.12. Before that he observed temperature in the morning with an ordinary
thermometer.  If the minimum thermometer was set at the evening observation the notes in the
column for morning observation should always be equal or lower than the evening
temperature the previous day. In December this is not true for 26% of the observations and in
June for 6%. These figures reduce to 6% and 2% in December and June respectively for
violations no more than 1°C. In practice different exposure of the two thermometers may
violate this logical test, and one should also take into account the possibility of instrumental



errors in Esmark's thermometers. We may conclude that the percentage of violation is not
large enough to contradict our conclusion that a night minimum thermometer was in use.

**4.2 The shift in 1821**
An inhomogeneity in the evening observation was detected by the homogeneity testing. It was
adjusted for by the mean difference between the midday observation and the evening
observation on each side of the shift, cf. Methods. The adjustments terms are presented in
Table 5. The adjustments are largest in the months where the daily temperature wave in
largest, so it is much likely that one reason for the shift was an earlier evening observation
time before 1822. If so it seems that the observation was taken at least one hour earlier before
this shift. Strictly speaking we know Esmark's observation times only in 1833, so this result is
not in contradiction to metadata. Other factors than the observation times might as well have
been involved, as the adjustments in winter is too large to be due to observation time only.

**4.3 A shift in the 1830s**
significant in winter, was detected by the SNHT double shift as well as the single shift when
the time window for the test was 1828.03-1838.12. The shift has the character of a continuous
inhomogeneity (Fig. 7). The difference between the evening observation and the morning
observation increased quite steadily from 1831 to 1838, whereas it was constant during the
years 1829-1831. The explanation may be a change in the observation times. According to
Esmark (1833) his observation times were, see Metadata.
• Morning: 08:30 ChT = 08:43 CET = 7:43 UTC
• Midday: 15:30 ChT = 15:43 CET = 14:43 UTC
• Evening: 21:30 ChT = 21:43 CET = 20:43 UTC
ChT = Christiania time i.e. local time for Christiania (Oslo), CET = Central European
Time, UTC = Universal Time Coordinated.
These observation times were for the barometric pressure, but at midday and evening the
thermometer were read at the same time as the barometer, but Esmark does not explicitly say
that the morning thermometer was read at the same time as the barometer. He also use the
term "in the latest" years so we do not know from which year these observation times were
introduced or if he continued to use them also in the following years 1834-1838.
Our hypothesis is that Esmark has had another observation time for the temperature
observations in the morning than for the pressure observations. Pressure was observed inside



the house, but for the temperature observations he had to leave the house for his garden.
Esmark might originally have observed temperature and pressure at the same time also in the
morning, but with the introduction of the minimum thermometer he could have thought that
the observation time for the morning temperature was not important. In spring, summer and
autumn he obviously was right in his thinking as minimum temperature occurs earlier than at
the morning observation time (8:30 ChT), but in winter the minimum temperature occurs
often later in the day as the systematic daily temperature wave is weak. This can explain the
changing difference during winter and the stable differences during the other seasons. As
Esmark grew older he might have gone outside for carrying out the morning observation later
and later. This might explain the trend shift in the morning observation. Following this
hypothesis the minimum temperature was adjusted, $\Delta T$, by use of formula (2) for the winter
season in accordance with the regression line shown in Fig. 7, where a = year (period 1832-
1838). No adjustments were undertaken for the period 1829-1831.

$$\Delta T = 0.2861 \cdot a - 523.85 \tag{2}$$

### 4.4 Overheating of the midday observation

The midday observation turned out to be homogenous, but it might have been overheated by
insufficient radiation protection in Esmark's yard. This was tested by comparison with the
Oslo – Blindern station that is well protected by a Stevenson screen. Difference between the
midday observation and the evening observation reveals a characteristic pattern (Fig. 8).
Whereas the differences were almost equal in the months September – March, the differences
in the Esmark series were larger than the differences in the Blindern series for the months
April – August. They were particularly large in MJA where the sun is highest on the sky and
the radiation reaches its annual maximum. Therefore our interpretation is that Esmark's
thermometer was overheated at the midday observation by (reflected) short wave radiation in
the period April – August, but not for the rest of the year. Based on the differences between
the two curves the adjustments of the midday observation are also given (lower panel in Fig.

8).


### 4.5 Homogenisation of the monthly mean temperature.

Esmark observed only three times a day, so it is far from obvious how monthly mean
temperature should be calculated without bias. This problem confronts meteorological
institutes worldwide so formulas for the calculation are developed (see Appendix B). The




formulas contain specific constants valid for each month and site. Strictly speaking the
constants were unknown for Esmark's observation site at Øvre Vollgate, but well known for
the station 18700 Oslo – Blindern lying 3.4 km to the north of Esmark's site. Fortunately
there are indications that the constants for Blindern could be used also for Øvre Vollgate (see
Appendix 2). Given the constants the calculation of homogenous monthly mean temperature
was trivial when the homogenised version of the observations at fixed hours was used. We
found that the adjustments for seasonal means vary from -0.7°C to +0.3°C (Fig. 9). The
adjustments were negative except from the last part of the series in winter and autumn. For the
annual means the adjustments are much less, they vary from -0.4°C to -0.1°C.

**4.6 The climate in Esmark's period of observation, 1816-1838**

Esmark's observations exhibit a long-term variation pattern characterised by lower values in
the start and in the end of the period, whereas the middle of the period was somewhat warmer,
cf. Fig. 10. This is true not only for the annual means, but also for all seasons of the year. For
individual years 1822 is warmest except in summer. The coldest year is 1816 followed by the
years 1817, 1820 and the last one 1838. In the year 1816 stands out as coldest also in two
seasons, spring (MAM) and autumn (SON), and also in the two individual months March and
May (not shown).

The year 1816 is of particular interest as it has gone into history as "the year without

summer" (Fagan, 2001). However, Esmark's observations show that this summer (JJA) was
not very extraordinary in Oslo, as the following summer of 1817 was colder, and in particular
that of 1821. More extraordinary is the spring temperature in 1816, being the only one with
mean temperature below zero. For agriculture the first years of Esmark's period of
observation must have been bad taking into account that low temperature is a limiting factor.
For the grain growing months (AMJJA) the mean temperature was about 8.5°C in the three
consecutive years 1816, 1817 and 1818, i.e. the lowest temperatures in Esmark series of
observation.

**5 Discussion**

From 1816 to the mid-1820s the annual Christiania temperature as recorded by Esmark rose
by approximately 1.5°C, then subsequenly slowly fell by almost 1°C towards 1840 (Fig. 10).
This general pattern is consistent with that found for the same time interval in the Swedish
capital Stockholm (compare with Fig. 5 in Moberg et al., 2002).



### 5.1 Adjusting for inhomogeneities


An important inhomogeneity was detected in Esmark's data at the end of 1822 in the evening
observation, and was adjusted for. Alternatively the inhomogeneity could be considered only
as a change of observational time, and not adjusted for by the testing. The series of mean
temperatures could then have been kept homogenous by assessing how much the observation
time had changed, leading to a corresponding change in the constants in Føyn's formula for
calculation of monthly mean temperature (see Appendix B). Probably also other changes
could have taken place at the end of 1822, so therefore we considered it better to apply the
adjustments directly to the temperature data, and use the same constants on both sides of the
shift for mean monthly temperature calculation. Moreover, there is some indication that a
changed environment could have played a role for this inhomogeneity as Esmark in 1823
planted fruit trees in his garden, cf. Metadata.
No doubt Esmark possessed a minimum thermometer from 1828. Such instruments were
available even before Esmark started his Oslo series in 1816. Already in 1790 a spirit
thermometer with a glass index, very much like those used up to this day at manual stations,
was described to the Royal Society in Edinburgh (Middleton, 1966: p. 152). In our work the
change from an ordinary thermometer reading at the morning observation to a minimum
thermometer reading was accounted for by a change of formula for mean monthly
temperature calculation. Therefore the series of mean monthly temperatures was kept
homogenous without adjusting for this shift in the morning observation.
The size of the adjustments of Esmark's observations gives an indication of the uncertainty of
Esmark's observations ( Fig. 9). These are adjustments for both homogeneity errors and short
wave radiation errors. They are largest during summer, which also are expected due to the
lack of radiation screens other than the wall of houses. For annual mean temperature the
adjustments are within the interval [-0.4°C, -0.1°C]. For individual observation times the
adjustments were higher [-0.7°C, +0.3°C].

### 5.2 Comparison with other observations


During the period 1822.11-1827.02  the Christiania professor  Christopher Hansteen carried
out observations at his home in Pilestredet at the corner of Keysersgate (Hansteen 1823, 1824,
1828; Birkeland, 1926: p. 12). The distance from Esmark's site was only about 600 m.
Hansteen's observation times varied much but for each month he gives the observation times
together with the data (Hansteen, 1824). The distribution of the observation times in UTC is
as follows: morning $06^h$ 4%, $07^h$ 44%, $08^h$ 52%; midday $13^h$ 20%, $14^h$ 78%, $15^h$ 2%; evening



$21^h$ 6%, $22^h$ 88%, $23^h$ 6%. Hansteen's observations were adjusted to Esmark's observation
times, approximately 08, 15 and 21 UTC by use of the mean daily temperature wave at
Blindern so that Esmark's observations could be compared with the adjusted ones of
Hansteen, Fig 11. It is evident that Hansteen's morning observation is much warmer than that
of Esmark except during winter. Much likely the thermometers of Hansteen had been
overheated as his two thermometers hang at the southern and northern side of the house
(Birkeland, 1925: 12). Then it must have been difficult to find shadow in the morning. Also
the midday observation is warmer by Hansteen than by Esmark. This is harder to understand.
If Birkeland's account of the thermometer at the north wall of the house is correct the house is
expected to give sufficient protection of that thermometer (Nordli et al., 2015), but as nothing
is known about the environment other factors might have been involved.
The evening temperature, however, is much in agreement with that of Esmark during summer
unlike for the two other observation times. The evening observations occurred after sunset at
both sites, whereas the two other observations occurred after sunrise. This supports the
suggestion that the differences at the morning and midday observations are due to radiation
errors.

Unlike the situation during summer, Hansteen's temperatures are lower than those of Esmark
in the period November – March (Fig. 11). In many weather situations the air loses energy by
long wave radiation because the short wave radiation is too small to compensate for the loss.
The result is that the coldest air is found at the lowest places in the local terrain, not
necessarily at the lowest sites above sea level. Esmark's house lies high in the local terrain at
the edge of a slope down to Pipervika cf. Metadata, whereas Hansteen's house lies low in the
local terrain at a floor of a small valley. The difference in winter temperature is therefore must
likely due to different local climate.

At The Astronomical Observatory in Oslo meteorological observations started in April 1837
that lasted almost for one hundred years  (Nordli et al., 2015), so this series overlaps Esmark's
series by 21 months. For comparison of the two series we have used unadjusted observations
from the observatory, whereas both adjusted and unadjusted Esmark observations are used
(Fig. 12). It is evident that for all seasons but winter Esmark's temperatures are lower than
those from the Observatory. Esmark died on 26 January 1839 (see  Metadata), so probably the
quality of the latest months of his series may be questioned. However, we cannot see any
declined quality directly from is observation protocols, but it is possible that the last two years



of his observations are not representative for Esmark's observational practice. Moreover, the
overlapping period is very short; only two years for most of the months, and only one year for
the months January to March. It is therefore possible that the present comparison is not valid
for Esmark's entire period of observation.

**5.3 The accuracy of the thermometers**
In Esmark's protocol for 1816 some instrumental corrections are given for what is claimed to
be Esmark's thermometer, Table 6. They are not written by Esmark himself, most probably
they are notes written by Birkeland, who says he has them after Hansteen 1821-23, but it is
not certain that they belong to the thermometer used by Esmark. The corrections are very
small for the frequent winter temperatures, but as high as 0.5°C for frequent summer
temperatures. Due to the uncertainty with the identification of Esmark's thermometer we have
not applied the corrections to his observations. It should also be kept in mind that Esmark
used another thermometer, i.e. a minimum thermometer for the period 1828.03-1838.12,
which might also have instrumental corrections. However, Esmark was a skilled instrument
builder, so it is not likely that he used thermometer with larger corrections that those in Table

6.


There were several volcanic eruptions that affected the world climate in the first years of
Esmark's period of observation. The Tambora eruption in 1815 was probably the greatest one.
It has given rise to the paradigm for 1816: "the year without a summer". Esmark's
observations show, however, that albeit being cold the summer was not extraordinary cold in
Oslo. And in the Stockholm series ("Bolin Centre Database,") the summer of 1816 was rather
warm, No 17 of the 23 summers from 1816-1838, ranged from low to high (Table 7). May,
however, was very cold in both cities, and July was quite warm in both cities, but in June and
August Oslo was much colder relative to the mean value than Stockholm.

There exist climate reconstructions for the period 1816-1838, independent from Esmark's
observations, based upon ice loss from Lake Randsfjorden (Nordli et al., 2007) temperature
proxy for the season February-April, and upon the date of grain harvest for Austlandet
(Nordli, 2001a), Vestlandet (Nordli et al., 2003), Lesja (Nordli, 2001b) and Trøndelag
(Nordli, 2004) temperature proxies for the seasons April-August and May-August (Table 8).
The three reconstructions within the county of South-Eastern Norway are all in agreement
with Esmark's observations that the summer of 1816 was among the coldest in the grain



growing seasons, whereas the reconstructions for the two other counties, Western and Mid
Norway, show relatively warm summers, even more so than those in Stockholm.

Anomalies of surface temperature and precipitation for the summer months of 1816 has been
reconstructed (Luterbacher and Pfister, 2015). They show a positive gradient from a cold core
of air lying over France towards Eastern and Northern Europe, so the paradigm of the severe
summer of 1816 has to be modified when it comes to Scandinavia and Eastern Europe. It
looks like this is easy to forget, e.g. "…weather patterns were disrupted worldwide for
months, allowing for excessive rain, frost, and snowfall through much of the Northeastern
U.S. and Europe in the summer of 1816"(Klingaman and Klingaman, 2014). It is therefore
important that the temperature gradient is recognised. The results in Table 8 are a part of the
pattern showing the spatial variability in Europe that summer.
**6 Conclusions**
Esmark's observations are almost complete for the years 1816-1838. Homogeneity testing
revealed a shift in the evening observation at the end of 1822. From March 1828 Esmark
noted nightly minimum temperature instead his previous notation of morning temperature.
During the years 1831 to 1838 the nightly minimum temperature increased almost steadily in
the winter season, i.e. it was inhomogenous. The homogenized temperature series showed low
temperature in both ends of the series, with higher temperature in the middle, i.s. the 1820s.
The starting year, 1816, is of particular interest as it has been referred to as the year without a
summer. The summer in Oslo was cold, but not extraordinary cold, as it was only the third
coldest in the period of observation. However, the annual mean of 1816 and also the months
March and May that year were the coldest ones in that period.








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



**Figure texts**
Fig. 1. Map of Christiania (now Oslo) 1811 with the location (red star) of
Esmark's house in Øvre Vollgt. 7 marked.
Fig. 2. Matriculation and survey 1830 of Esmark's property No. 308, Øvre Voldgate 7, in
Oslo Byarkiv (City archives). Arrow indicates N. Garden to the left, house surrsounding back
yard to the right.
Fig. 3. Street view of Esmark's house in Øvre Voldgate 7. Photograph from around 1900.
Oslo Bymuseum, No. OB.F00897. High buildings on each side built late 19[th] century.
Fig. 4. Esmark's Christiania protocol for 1817. Now deposited at Riksarkivet
(National archives), Oslo. S-1570. Det norske meteorologiske institutt. F/Fa.
Materiale etter professorer. L0002.
Fig. 5. The January page from Esmark's meteorological observation protocol from
1823, the year he discovered ice ages. Now deposited at Riksarkivet (National
archives), Oslo. S-1570. Det norske meteorologiske institutt. F/Fa. Materiale etter
professorer. L0002.
Fig. 6. The first published Christiania weather table, from *Den norske Rigstidende*,
24 January 1818.
Fig. 7 The temperature difference (ºC) between Esmark's evening observation and the
morning observation the following day for the winter season (Dec-Feb).
Fig. 8 Temperature differences (ºC) between the observations at Blindern at 15 UTC and at 21
UTC for the period 1993.01-2015.09. Also the difference between the midday and evening
observations of Esmark is shown for the period 1816.01-1838.12. (The adjustments of the
evening observations, Table 5, are added to the data for the period 1816.01-1821.12 before
the calculation of the differences. In the table below the figure are shown the adjustments of
Esmark's midday observation
Fig. 9. Adjustments added to Esmark's series for each season during his period of
observation, 1816-1838.
Fig. 10. Annual and seasonal means of Esmark's temperature series (symbols), and Gaussian
filter (curves) with standard deviation 3 in the Gaussian distribution (e.g. Nordli et al., 2015),
corresponsing roughly to a 10 year regtangular filter.



Fig. 11. Difference between Esmark's observations at Øvre Vollgate and Hansteen's
observations at Pilestredet (Esmark minus Hansteen) during the period 1822.11-1827.02 at
08, 15 and 21 UTC. The monthly means are calculated by Føyn's formula, cf. Appendix 1

Fig. 12. Differences in mean monthly temperature between Esmark's observations at Øvre
Vollgate and those at the Astronomical Observatory (Esmark minus Observatory) during the
period 1837.04-1838.12. Esmark's observations are presented both unadjusted and adjusted.
For the observatory the temperatures are unadjusted.











**Tables**



Table 1 Esmark's station at Øvre Vollgate 7 as well as other observation stations used in this article:
national station number (identifier) and name, period of observation and station altitude. The star
before the start year marks the start of hourly observations

| No. and name | Period (from-to; year, month, day | $H_s$ (m) |
|---|---|---|
| 18651 Oslo II | 1837.04.02-1933.12.31 | 25 |
| 18654 Oslo - Øvre Vollgate | 1816.01.01-1838.12.31 | 11 |
| 18655 Oslo - Pilestredet | 1822.10.19-1827.02.28 | 16 |
| 18700 Oslo - Blindern | *1993.01.05 to present | 94 |
| 18815 Oslo - Bygdøy | *2012.01.01 to present | 15 |



Table 2 The SNHT test used for comparison of observations at time x versus observations at time y (x
vs y). The shifts are given by the last year of each part of the series. For the single shift test also the
adjustment needed for the x-series to be homogenous with y-series (Non-significant results are given
in italic).

| Part 1, 1816.01-1838.12: The whole length of the series | | | | | | |
|---|---|---|---|---|---|---|
| SNHT tests | Obs. times | Winter | Spring | Summer | Autumn | Year |
| Single shift | I vs II | 1833; -1.1 | 1827; -2.1 | 1827; -3.3 | 1824; -1.4 | 1827; -1.8 |
| Single shift | I vs III | 1832; -1.5 | 1826; -2.8 | 1827; -4.0 | 1827; -1.7 | 1827; -2.4 |
| Single shift | III vs II | 1821; 0.7 | 1820; 1.5 | 1821; 1.3 | *1821; 0.6* | 1821; 0.9 |
| Double shift | I vs II | 1826; 1834 | 1818; 1827 | 1817; 1827 | 1824; 1829 | 1823; 1827 |
| Double shift | I vs III | 1819; 1832 | 1820; 1826 | 1818; 1828 | 1823; 1829 | 1818; 1827 |
| Double shift | III vs II | *1821; 1832* | 1819; 1835 | 1821; 1835 | *1817; 1834* | 1821; 1835 |
| Part 2, 1816.01 – 1828.02 | | | | | | |
| SNHT-tests | Obs. times | Winter | Spring | Summer | Autumn | Year |
| Single shift | I / II | *1826; -0.8* | *1818; -0.7* | *1817; -0.8* | 1824; -1.0 | *1823; -0.5* |
| Single shift | I /III | 1818; -1.0 | 1820; -1.7 | 1818; -1.7 | 1821; -0.9 | 1818; -1.3 |
| Single shift | III / II | *1821; 0.6* | 1819; 1.4 | *1821; 1.2* | *1817; 0.8* | 1821; 0.8 |
| Part 3, 1828.03 – 1838.12 | | | | | | |
| SNHT-tests | Obs. times | Winter | Spring | Summer | Autumn | Year |
| Single shift | I / II | 1834; -1.0 | *1834; 0.4* | *1830; -0.4* | 1829; -0.4 | *1830; -0.5* |
| Single shift | I /III | 1832; -1.3 | *1836; -0.6* | *1836; -0.8* | 1829; -0.9 | *1836; -0.8* |
| Single shift | III / II | *1833; 0.4* | 1835; 0.8 | *1835; 0.9* | 1834; 0.6 | *1835; 0.7* |



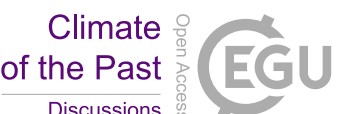

Table 3. The same as Table 1, but the single shift test used on monthly resolution. In the 1[st] and 3[rd]
rows are given the year of the shifts, and in the 2[nd] and 4[th] rows the adjustments.

|  | Jan | Feb | Mar | Apr | May | Jun | Jul | Aug | Sep | Oct | Nov | Dec |
|---|---|---|---|---|---|---|---|---|---|---|---|---|
| I/II | 1834 | 1826 | *1826* | 1830 | 1827 | 1827 | 1827 | 1827 | 1825 | 1827 | 1824 | 1833 |
|  | -1.2 | -1.4 | *-1.0* | -2.2 | -3.3 | -3.4 | -3.5 | -2.9 | -1.9 | -1.1 | -1.5 | -1.2 |
| III/II | 1828 | *1832* | 1820 | 1819 | 1819 | 1826 | *1821* | 1821 | *1821* | 1820 | *1834* | 1820 |
|  | 0.6 | *0.7* | 1.1 | 1.7 | 1.8 | 1.3 | *1.3* | 1.3 | *0.8* | 0.9 | *0.6* | 0.7 |


Table 4 Difference, Diff (°C), of median temperature between Esmark's evening observations and the
observations the following morning during different time intervals. The similar differences for the
modern station Oslo – Blindern are also shown, i.e. the observation at 21 UTC and the minimum
temperature at 08 UTC. Also the standard deviations, STD (°C), of the differences are shown.

|  |  | Jan | Feb | Mar | Apr | May | Jun | Jul | Aug | Sep | Oct | Nov | Dec |
|---|---|---|---|---|---|---|---|---|---|---|---|---|---|
| Esmark | Diff | 0.0 | 0.0 | 0.0 | -0.7 | -1.8 | -1.6 | -1.3 | -1.2 | 0.0 | 0.5 | 0.0 | 0.0 |
| 1816.01-1821.12 | STD | 3.4 | 2.6 | 2.4 | 2.1 | 2.4 | 2.3 | 2.6 | 2.1 | 2.1 | 2.0 | 2.6 | 2.2 |
| Esmark | Diff | 0.9 | 0.7 | 1.2 | 0.6 | 0.6 | -0.7 | -0.6 | 0.0 | 1.2 | 0.6 | 0.8 | 0.6 |
| 1822.01-1828.02 | STD | 3.1 | 2.5 | 2.3 | 1.8 | 2.2 | 2.4 | 2.2 | 2.1 | 2.9 | 2.5 | 2.5 | 2.4 |
| Esmark | Diff | 1.3 | 1.5 | 1.9 | 2.2 | 3.1 | 3.1 | 3.1 | 3.1 | 2.5 | 1.9 | 1.6 | 1.3 |
| 1828.03-1838.12 | STD | 2.6 | 2.3 | 2.5 | 1.8 | 2.1 | 2.2 | 2.4 | 2.3 | 2.2 | 2.1 | 1.9 | 2.7 |
| Blindern | Diff | 1.0 | 1.5 | 2.3 | 2.6 | 3.2 | 3.0 | 2.7 | 2.4 | 2.0 | 1.5 | 1.0 | 1.0 |
| 1993.09-2015.09 | STD | 1.7 | 1.8 | 1.8 | 1.7 | 1.8 | 1.8 | 1.7 | 1.6 | 1.6 | 1.6 | 1.5 | 1.6 |


Table 5 Adjustment (°C) of the evening observation in the period 1816.01-1821.12

| Jan | Feb | Mar | Apr | May | Jun | Jul | Aug | Sep | Oct | Nov | Dec |
|---|---|---|---|---|---|---|---|---|---|---|---|---|
| -0.6 | -0.6 | -1.0 | -1.2 | -1.3 | -1.2 | -1.3 | -1.3 | -0.9 | -0.8 | -0.3 | -0.5 |


Table 6. Instrument correction (Corr) for thermometer readings (Temp.). The thermometer might have
been used by Esmark, 1816-1838.

| Temp. (°C) | 25.00 | 18.75 | 12.50 | 6.25 | 0.00 | -6.25 | -12.50 | -18.75 | -25.00 |
|---|---|---|---|---|---|---|---|---|---|
| Corr. (°C) | +0.50 | +0.50 | +0.38 | +0.38 | +0.13 | +0.13 | +0.13 | +0.13 | +0.63 |


Table 7 The range of mean temperature in 1816 for months and seasons during the years 1816-1838
for Oslo (Esmark's observations). For comparison also Stockholm is included. The range runs from
low to high values.

|  | J | F | M | A | M | J | J | A | S | O | N | D | Yr | Wi | Sp | Su | Au |
|---|---|---|---|---|---|---|---|---|---|---|---|---|---|---|---|---|---|
| Oslo | 14 | 5 | 1 | 3 | 1 | 3 | 12 | 2 | 2 | 3 | 7 | 10 | 1 | 9 | 1 | 3 | 2 |
| Stockholm | 14 | 3 | 6 | 9 | 1 | 16 | 18 | 9 | 13 | 5 | 8 | 12 | 7 | 6 | 4 | 17 | 3 |




Table 8 The range of mean temperature in 1816 for seasons during the years 1816-1838 for Oslo
(Esmark's observations), and for climate reconstructions from proxy data at different places in Norway.
For comparison also Stockholm is included. The range runs from low to high values.

| Place, County | February - April | April – August | May-August |
|---|---|---|---|
| Oslo, South-eastern Norway | 2 | 1 | 3 |
| Randsfjorden, South-eastern Norway | 2 | | |
| Austlandet, South Eastern Norway | | 1 | |
| Lesja, South-eastern Norway | | | 1 |
| Bergen, Western Norway | | 18 | |
| Trøndelag, Mid Norway | | | 18 |
| Stockholm, Sweden | 3 | 10 | 9 |





**APPENDIX A. ESMARK'S METEOROLOGICAL TABLES IN**
*DEN NORSKE RIGSTIDENDE.*

Esmark, J. 1818/19. Meterologiske Iagttagelser i Christiania 1818, anstillede af
Prof. Esmark. *Den Norske Rigstidende* 1818, No. 7 (24 January); No. 10 (4
February); No. 14 (18 February); No. 18 (4 March); No. 23 (21 March), No.
28 (8 April), No. 32 (22 April); No. 37 (9 May); No. 40 (20 May), No. 45 (6
June), No. 49 (20 June), No. 54 (8 July); No. 59 (25 July); No. 63 (8
August); No. 67 (21 August); No. 71 (5 September); No. 83, (17 October);
No. 84 (21 October), No. 86 (28 October); No. 88 (4 November); No. 95 (28
November); No. 98 (9 December); No. 102 (23 December); No. 3 (8 January

1819).

Esmark, J. 1819/20. Meterologiske Iagttagelser i Christiania 1819, anstillede af
Prof. Esmark. *Den Norske Rigstidende* No. 6 (19 January); No. 11 (5
February); No. 16 (23 February); No. 19 (5 March); No. 24 (23 March); No.
(6 April); No. 33 (23 April); No. 36 (4 May); No. 41 (21 May); No. 48
(15 June); No. 49 (18 June); No. 54 (6 July); No. 62 (3 August); No. 65 (13
August); No. 67 (20 August); No. 78 (28 September); No. 79 (1 October)
No. 82 (12 October); No. 84 (19 October); No. 89 (5 November); No. 95 (26
November); No. 99 (10 December); No. 103 (24 December); No. 2 (7
January 1820).
Esmark, J. 1820/21. Meterologiske Iagttagelser i Christiania 1820, anstillede af
Prof. Esmark. *Den Norske Rigstidende*, No. 7 (25 January); No. 11 (8
February), No. 14 (18 February); No. 18 (3 March); No. 24 (24 March) ; No.
(7 April); No. 32 (21 April); No. 37 (9 May); No. 41 (23 May); No. 47
(13 June); No. 50 (23 June); No. 54 (7 July); No. 58 (21 July); No. 63 (8
August); No. 68 (25 August); No. 72 (8 September); No. 77 (26 September);
No. 81 (10 October); No. 85 (24 October); No. 88 (3 November); No. 94 (24
November); No. 98 (8 December); No. 103 (26 December); No. 3 (9 January

1821).

Esmark, J. 1821/22. Meterologiske Iagttagelser i Christiania 1821, anstillede af
Professor Esmark. *Den Norske Rigstidende*, No. 7 (23 January), står bare
snee,men ikke mengde, ; No. 11 (6 February); No. 16 (23 February); No. 21
(13 March); No. 23 (20 March); No. 29 (10 April); No. 33 (24 April), No. 38





(11 May); No. 41 (22 May); No. 45 (5 June); No. 52 (29 June); No. 55 (10
July); No. 58 (20 July); No. 63 (6 August); No. 68 (24 August); No. 72 (7
September); No. 76 (21 September); No. 80 (5 October); No. 85 (22
October); No. 89 (5 November); No. 93 (19 November)(nytt moderne
plusstegn); No. 98 (7 December); No. 102 (21 December); No. 2 (7 January
1822).

Esmark, Jens 1822/23. Meteorologiske Iagttagelser i Christiania 1822, anstillede
ved Professor Esmark. *Den Norske Rigstidende*, No. 5 (18 January); No. 10
(4 February); No. 15 (22 February); No. 18 (4 March); No. 23 (22 March);
No. 28 (8 April); No. 32 (22 April); No. 36 (6 May); No. 42 (27 May); No.
(7 June) not nedbørmåling; No. 50 (24 June); No. 81 (11 October); No. 82
(14 October); No. 83 (18 October); No. 84 (21 October); No. 87 (1
November); No. 89 (8November); No. 90 (11 November); No. 92 (18
November); No. 94 (25 November); No. 96 (2 December); No. 98 (9
December); No. 102 (23 December); No. 2 (6 January 1823).
Esmark, J. 1823/24. Meteorologiske Iagttagelser i Christiania 1823, anstillede ved
Professor Esmark. *Den Norske Rigstidende* No. 7 (24 January); No. 11 (7
February) ; No. 15 (21 February); No. 20 (10 March); No. 24 (24 March);
No. 27 (4 April); No. 31 (18 April); No. 36 (5 May); No. 40 (19 May); No.
(9 June); No. 49 (20 June); No. 75 (19 September); No. 76 (22
September); No. 77 (26 September); No. 78 (29 September); No. 79 (3
October); No. 81 (10 October); No. 82 (13 October); No. 84 (20 October);
No. 88 (3 November); No. 93 (21 November); No. 98 (8 December); No. 102
(22 December); No. 2 (5 January 1824).
Esmark, J. 1824/25. Meteorologiske Iagttagelser i Christiania 1824, anstillede ved
Professor Esmark. *Den Norske Rigstidende* No. 6 (19 January); No. 11 (5
February); No. 15 (19 February); No. 20 (8 March); No. 24 (22 March); No.
29 (8 April); No. 33 (22 April); No. 37 (6 May); No. 42 (24 May); No. 45 (3
June); No. 50 (21 June); No. 54 (5 July); No. 59 (22 July);  No. 64 (9
August); No. 68 (23 August); No. 74 (13 September); No. 77 (23
September); No. 80 (4 October); No. 86 (25 Oktober); No. 89 (4 November);
No. 96 (29 November); No. 98 (6 December); No. 103 (23 December); No. 2
(6 Januar 1825).



Esmark, J. 1825/26. Meteorologiske Iagttagelser i Christiania 1825, anstillede ved
Professor Esmark. *Den Norske Rigstidende* No. 7 (24 January); No. 11 (7.
February), No. 15 (21 February); No. 18 (3. March); No. 24 (24 March); No.
29 (11 April); No. 33 (25 April); No. 36 (5 May); No. 40 (19 May); No. 45
(6 June); No. 49 (20 June); No. 53 (4 July); No. 70 (1 September); No. 71 (5
September); No. 73 (12 September); No. 74 (15. September); No. 76 (22
September); No. 79 (3 October), No. 85 (24 October); No. 89 (7 November);
No. 93 (21 November); No. 97 (5 December); No. 102 (22 December); No. 2
(5 January 1826).
Esmark, J. 1826/27. Meteorologiske Iagttagelser i Christiania 1826, anstillede ved
Professor Esmark. *Den Norske Rigstidende* No.8 (26 January); No. 12 (9
February); No. 17 (27 February); No. 19 (6 March); No.23 (20 March); No.
28 (6 April); No. 33 (24 April); No. 36 (4 May); No. 43 (29 May); No. 45 (5
June);  No. 50 (22 June); No. 55 (10 July): No.58 (20 July); No. 62 (3
August); No. 67 (21 August); No. 72 (7 September); No. 77 (25 September);
No. 80 (5 Oktober); No. 84 (19 October); No. 88 (2 November); No. 93 (20
November); No. 97 (4 December); No. 102 (21 December); No. 2 (4 January
1827).
Esmark, J. 1827/28. Meteorologiske Iagttagelser i Christiania 1827, anstillede ved
Professor Esmark. *Den Norske Rigstidende ,* No. 7 (22 January); No. 11 (5
February); No. 16 (22 February); No. 19 (5 March); No. 24 (22 March); No.
28 (5 April); No. 32 (19 April); No. 37 (7 May); No. 43 (28 May); No. 48
(14 June); No. 50 (21 June); No. 54 (5 July); No. 58 (19 July); No. 79 (1
October); No. 80 (4 October); No. 81 (8 October); No. 82 (11 October); No.
(15 October); No. 84 (18 October); No. 89 (5 November); No. 94 (22
November); No. 97 (3 December); 102 (20 December); No. 2 (7 January
1828) – also sums up last ten years, compares with Stockholm, the coldest
982        years have been 1819 and 1820, the mildest 1822 and 1826.
Esmark, J. 1828/29. Meteorologiske Iagttagelser i Christiania 1828, anstillede ved
Professor Esmark. *Den Norske Rigstidende ,* No. 6 (21 January); No. 10 (4
February); No. 15 (21 February); No. 18 (3 March); No. 24 (24 March); No.
27 (3 April – mange solpletter); No. 32 (21 April); No. 36 (5 May); No. 40
(19 May); No. 45 (5 June); No. 49 (19 June); No. 53 (3 July); No. 59 (24
July); No. 63 (7 August); No. 78 (29 September); No. 79 (2 October); No. 81





(9 October); No. 84 (20 October); No. 88 (3 November); No. 94 (24
November); No. 98 (8 December); No. 102 (22 December); No.2 (5 January

1829).

Esmark, J. 1829/30. Meteorologiske Iagttagelser i Christiania 1829, anstillede ved
Professor Esmark. *Den Norske Rigstidende* , No. 8 (26 January); No. 11 (5
February); No. 15 (19 February); No. 19 (5 March – den strengeste vinter på
mange år); No. 24 (23 March); No. 27 (2 April); No. 33 (23 April); No. 37 (7
May); No. 42 (25 May); No. 46 (8 June); No. 50 (22 June); No. 54 (6 July);
No. 78 (28 September); No. 79 (30 September); No. 80 (5 October); No. 81
(8 October); No. 85 (22 October); No. 87 (29 October); No. 89 (5
November); No. 90 (9 November); No. 94 (23 November); No. 99 (10
December); No. 103 (24 December); No. 2 (7 January 1830).
Esmark, J. 1830/31. Meteorologiske Iagttagelser i Christiania 1830, anstillede ved
Professor Esmark. *Den Norske Rigstidende,* No. 7 (25 January); No. 11 (8
February); No. 14 (18 February); No. 18 (4 March); No. 22 (18 March); No.
27 (5 April); No. 31 (19 April); No. 36 (6 May); No. 40 (19 May); No. 46 (9
June); No. 50 (23 June); No. 53 (5 July); No. 57 (19 July); No. 63 (9
August); No. 70 (1 September); No. 73 (13 September); No. 78 (29
Septmerber); No. 81 (11 October); No. 84 (21 October); No. 91 (15
November); No. 95 (29 November); 98 (9 December); No. 102 (23
December); No. 3 (10 January 1831*)*.
Esmark, J. 1831/32. Meteorologiske Iagttagelser i Christiania 1831, anstillede ved
Professor Esmark. *Den Norske Rigstidende* , No. 10 (3 February); No. 11 (7
February); No. 17 (28 February); No. 20 (10 March); No. 25 (28 March); No.
28 (7 April); No. 33 (25 April); No. 39 (12 May); No. 43 (22 May); No. 52
(12 June); No. 57 (23 June); No. 63 (7 July); No. 70 (24 July); No. 75 (4
August); No. 85 (28 August); No. 88 (4 September); No. 97 (25 September);
No. 102 (10 October); No. 110 (3 November); No. 112 (10 November); No.
118 (1 December); No. 119 (4 December); No. 1 (1 January 1832) ; No. 2 (5
January 1832).
Esmark, J. 1832/33. Meteorologiske Iagttagelser i Christiania 1832, anstillede ved
Professor Esmark. *Den Norske Rigstidende,* No.10 (2 February); No. 11 (5
February); No. 19 (4 March); No. 20 (8 March); No. 26 (26 March); No. 30
(12 April); No. 33 (22 April); No. 37 (6 May); No. 43 (20 May); No. 52 (10





Juni); No. 57 (21 Juni); No. 63 (5 July); No. 70 (22 July); No. 78 (9 August);
No. 86 (28 August – usedvanlig kold sommer); No. 92 (11 September); No.
(25 September); No. 103 (7 October); No. 108 (25 October); No. 111 (4
November); No. 117 (25 November); No. 122 (13 december); No. 127 (30
December); No.  4 (13 Januery 1833).
Esmark, J. 1833/34. Meteorologiske Iagttagelser i Christiania 1833, anstillede ved
Professor Esmark. *Den Norske Rigstidende,* No.10 (3 February); No. 12 (10
February); No. 18 (3 March); No. 24 (24 March); No. 25 (28 March); No. 30
(14 April); No. 35 (2 May); No. 37 (9 May); No. 44 (26 May); No. 50 (9
June); No. 58 (27 June); No. 63 (9 July); No. 77 (11 August); No. 80 (18
August); No. 86 (1 September); No. 91 (12 September); No. 97 (26
September); No. 103 (13 October); No. 105 (20 October); No. 110 (7
November); No. 115 (24 November); No.120 (12 December); No. 123 (22
December); No. 2 (5 January 1834).
Esmark, J. 1834/35. Meteorologiske Iagttagelser i Christiania 1834, anstillede ved
Professor Esmark. *Den Norske Rigstidende ,*No. 7 (23 Januery); No. 10 (2
February); No. 16 (23 February); No. 18 (2 March); No. 24 (23 March); No.
27 (3 April); No. 32 (20 April); No. 37 (4 May); No. 43 (18 May); No. 53
(10 June); No. 60 (26 June); No. 68 (15 July)(regnet som falt på en
kvadratfods flate utgjorde 4 rhinlandskae tommer eller 576 kubikktommer);
No. 71 (22 July); No. 79 (10 August), No. 83 (19 August); No. 90 (7
September); No. 96 (21 September); No. 102 (5 October); No. 107 (23
October); No. 111 (6 November); No. 117 (27 November); No. 119 (4
December); No. 126 (28 December); No. 2 (8 January 1835).
Esmark, J. 1835/36. Meteorologiske Iagttagelser i Christiania 1835, anstillede ved
Professor Esmark. *Den Norske Rigstidende,* No. 10 (1 February); No. 12 (8
February); No.15 (19 February); No. 20 (8 March); No. 24 (22 March); No.
28 (5 April); No. 34 (26 April); No. 40 (10 May); No. 50 (2 June); No. 54
(11 June); No. 58 (21 June); No. 65 (7 July); No. 72 (23 July); No. 79 (9
August); No. 88 (30 August); No. 91 (6 September); No. 99 (24 September);
No. 105 (11 October); No. 107 (18 October); No. 112 (5 November); No.
(26 November); No. 120 (3 December); No. 126 (24 December); No. 3
(10 January 1836).




Esmark, J. 1836/37. Meteorologiske Iagttagelser i Christiania 1836, anstillede ved
Professor Esmark. *Den Norske Rigstidende,* No. 7 (24 January); No. 15 (21
February); No. 17 (28 February); No. 19 (6 March); No. 23 (20 March); No.
27 (3 April); No. 32 (21 April); No. 38 (5 May); No. 45 (22 May); No. 50 (2
June); No. 59 (23 June); No. 66 (10 July); No. 70 (19 July); No. 78 (7
August); No. 85 (23 August?) ; No. 92 (8 September); No. 98 (22
September); No. 105 (9 October); No. 111 (30 October); No. 112 (3
November); No. 119 (27 November); No. 125 (18 December); No. 126 (22
December); No. 3 (5 January 1837).
Esmark, J. 1837/38. Meteorologiske Iagttagelser i Christiania 1837, anstillede ved
Professor Esmark. *Den Norske Rigstidende*, No. 10 (22 January); No. 17 (7
February); No. 22 (19 February); No. 22 (2 March); No. 34 (19 March); No
41 (4 April); No. 48 (20 April); No. 53 (2 May); No. 61 (21 May); No. 67 (4
June); No. 74 (20 June); No. 82 (9 July); No. 86 (18 July); No. 93 (3
August); No. 100 (20 August); No. 106 (3 September); No. 113 (19
September); No. 120 (5 October); No. 126 (19 October); No. 132 (2
November); No. 139 (19 November); No. 145 (3 December); No. 152 (19
December); No. 2 (4 January 1838).
Esmark, J. 1838. Meteorologiske Iagttagelser i Christiania 1838, anstillede ved
Professor Esmark. *Den Norske Rigstidende*, No. 10 (18 January); No. 19 (3
February); No. 29 (20 February); No. 36 (4 March); No. 45 (20 March); No.
53 (3 April); No. 62 (19 April); No. 70 (3 May); No. 79 (19 May); No. 87 (2
June); No. 98 (19 June); No. 108 (4 Junly); No. 117 (19 July); No. 127 (2
August); No. 137 (19 August); No. 148 (6 September); No. 156 (20
September); No. 164 (4 October); No. 173 (20 October); No. 181 (3
November); No. 190 (18 November); No. 199 (4 December); No. 207 (18
December).
**Appendix B**
MET Norway calculates monthly mean temperatures for manual stations by Føyn's and
Köppen's formulas (Birkeland, 1936; Gjelten et al., 2014; Nordli et al., 2015), so we chose to
use those formulas also for Esmark's observations: The monthly mean temperature, T, may be
calculated by Føyn's formula and a modified Köppen's formula, Table A1.



Table A1. Formulas for calculation of monthly mean temperature, T, where $T_{08}$, $T_{15}$ and $T_{21}$, are
monthly means at observation times 08, 15 and 21 UTC respectively, and $T_n$ is monthly mean night
temperature, $k_g$ and $k_f$ are constants.

| Føyn's formula | $T = T_g + k_g(T_{15} - T_g)$ | $T_g = \dfrac{T_{08} + T_{21}}{2}$ |
|---|---|---|
| Köppen's formula | $T = T_f - k(T_f - T_n)$ | $T_f = \dfrac{T_{15} + T_{21}}{2}$ |


A "true" monthly mean temperature, T, may be calculated by the arithmetic mean of hourly
observation according to definition, so for a station that have hourly observations the
constants, $k_g$ and $k_f$, are easily calculated by rearranging Føyn's and Köppen's formulas. For
Esmark's series from Øvre Vollgate the constants were unknown. It was assumed that the
constants from Blindern could be used also for Øvre Vollgate. An indication of the robustness
of this assumption was tested by comparison with a short series of hourly observations from
the station 18815 Oslo – Bygdøy, 15 m a.s.l. The test procedure started with calculation of the
constants for the Blindern series based on the period 2012.12-2015.09. These constants were
then used for the calculation of mean monthly temperatures for Bygdøy for the same period,
which were compared with the "true" monthly means, i.e. those calculated by the hourly
observations. For Føyn's formula the deviation from the true means varied from -0.06°C in
December to +0.18°C in March that gave +0.05°C for the whole year. Corresponding figures
for Köppen's formula were -0.06°C in July, +0.16°C in September and +0.01°C for the whole
year. These differences are so small that the lack of exact knowledge of the constants does
add practically no uncertainty to the monthly temperatures.

Fig. 12. Differences in mean monthly temperature between Esmark's observations
at Øvre Vollgate and those at the Astronomical Observatory (Esmark minus
Observatory) during the period 1837.04-1838.12. Esmark's observations are
presented both unadjusted and adjusted. For the observatory the temperatures
are unadjusted.





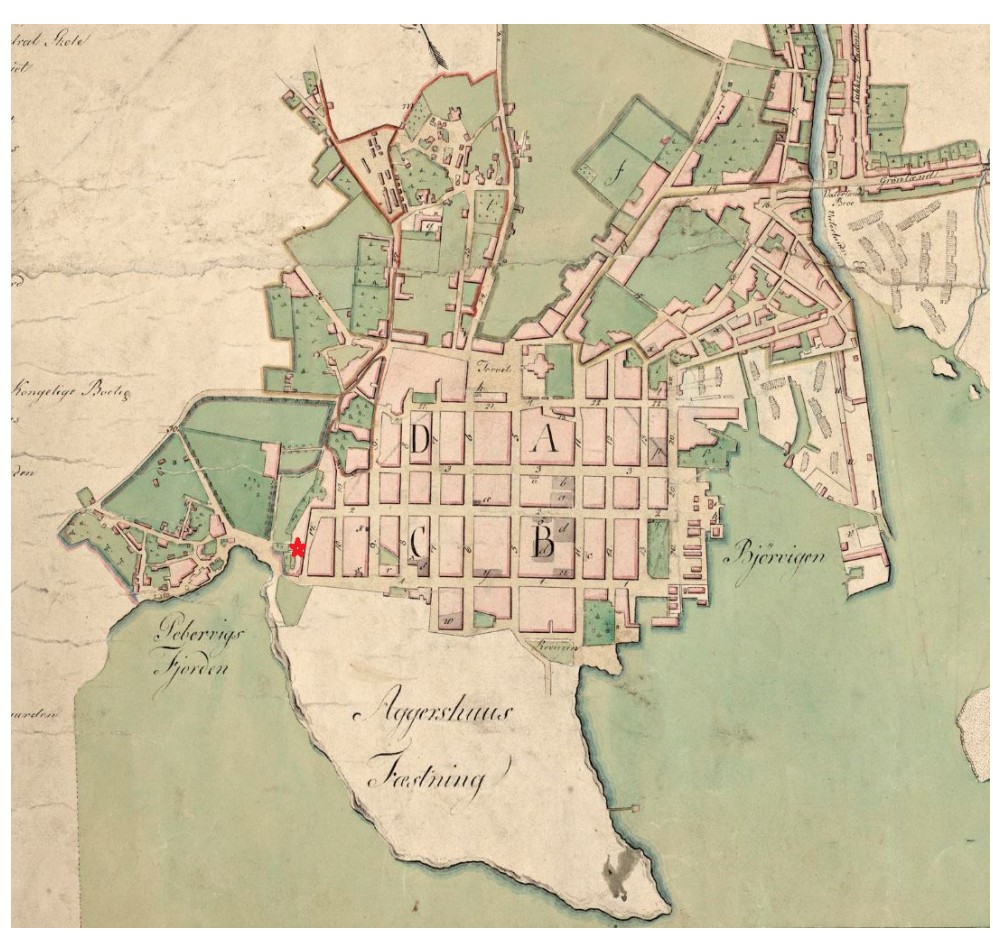

Fig. 1. Map of Christiania (now Oslo) 1811 with the location of Esmark's house in

Øvre Vollgt. 7 marked with red star.






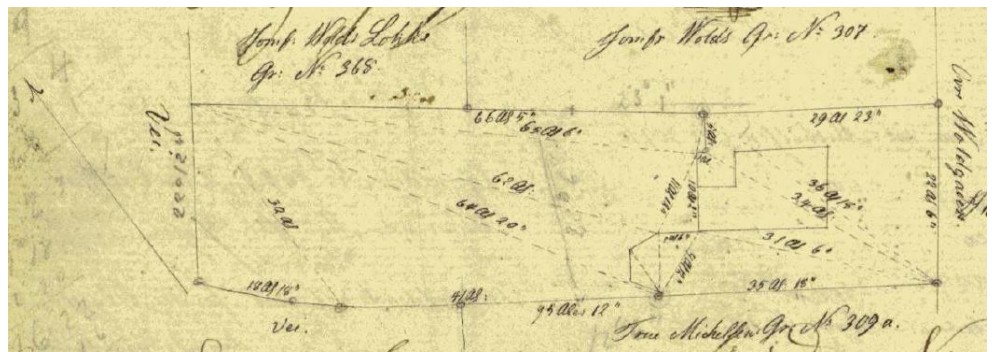

Fig. 2. Matriculation and survey 1830 of Esmark's property No. 308, Øvre Voldgate 7, in
Oslo Byarkiv (City archives). Garden to the left, house surrounding back yard to the right.

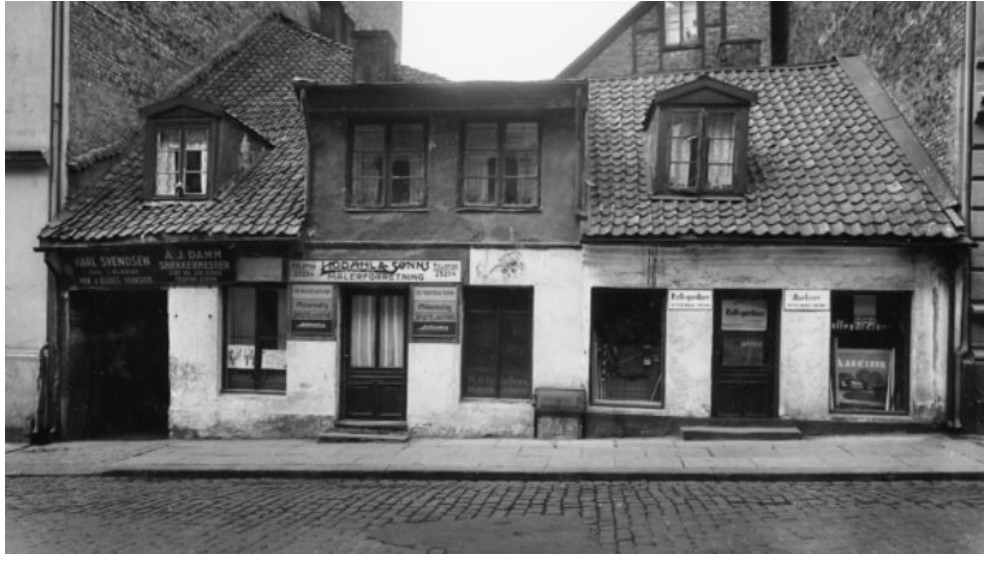

Fig. 3. Street view of Esmark's house in Øvre Voldgate 7. Photograph from around 1900. The
higher houses on both sides are late 19th century. Oslo Bymuseum, No. OB.F00897.



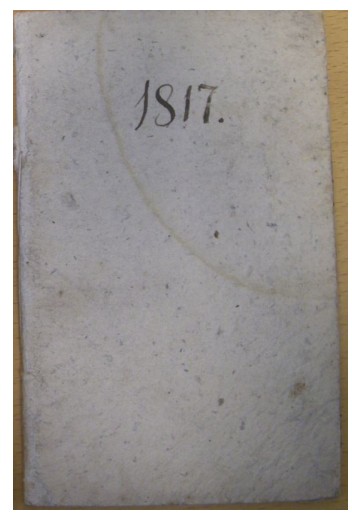

Fig. 4. Esmark's Christiania protocol for 1817. Now deposited at Riksarkivet

(National archives), Oslo. S-1570. Det norske meteorologiske institutt. F/Fa.

Materiale etter professorer. L0002.


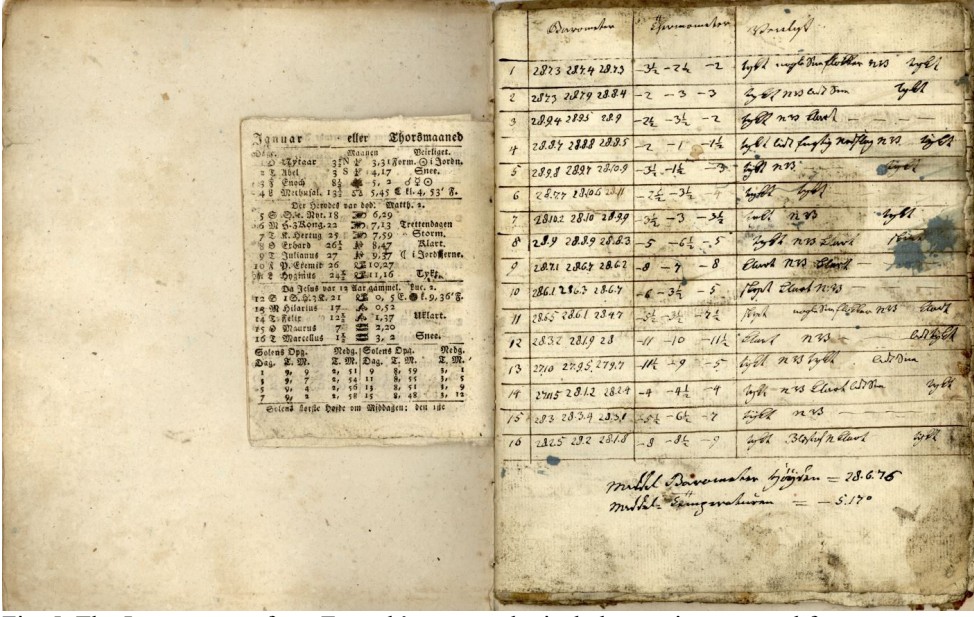

Fig. 5. The January page from Esmark's meteorological observation protocol from

1823, the year he discovered ice ages. Now deposited at Riksarkivet

(National archives), Oslo. S-1570. Det norske meteorologiske institutt. F/Fa.

Materiale etter professorer. L0002.




Meteorologiſke Jagttagelſer i Chriſtiania 1818,
anſtillede af Prof. Esmark.

| Januar. | Barometret. | | Thermom. | Veirliget. |
|---|---|---|---|---|
| 1 | 28T. | 3 L. | — 11⅕° | Taage og tykt Veir |
| 2 | 28 | 6¼ | — 10¼ | Skyet. |
| 3 | 28 | 6¾ | — 8⅔ | Tykt Veir. |
| 4 | 28 | 5 | — 11⅙ | Lidt Snee. |
| 5 | 28 | 1⅔ | — 9⅓ | Lidt Snee. |
| 6 | 27 | 11⅔ | — 4⅙ | Tykt og lidt Snee. |
| 7 | 27 | 6⅙ | �превот 3¾ | Tykt Veir. |
| 8 | 27 | 5⅙ | ✝ ¾ | Stærk Taage. |
| 9 | 27 | 10⅓ | — 4¼ | Taage. |
| 10 | 27 | 5¾ | ✝ 1¼ | Bl. af S., Nordlys |
| 11 | 27 | 6¼ | ✝ 1½ | Klart Veir. |
| 12 | 27 | 6¼ | ✝ ¼ | Sn. og Regn SV |
| 13 | 27 | 5⅙ | 0 | Sn. og Regn SV |
| 14 | 27 | 6⅓ | ✝ ½ | Klart. |
| 15 | 26 | 10⅓ | ✝ ½ | Snee og Bl. af S. |

Anmærkninger: Observationerne ere anſtille-
de 34 Rhinlandſke Fod over Havet, og ere Mid-
deltallet af Observationer, anſtillede Morgen,
Middag og Aften. Barometer-Høiderne ere cor-
rigerede ſaaledes, ſom de ſkulle være, derſom
Barometret havde været udſat for 0° Tempera-
tur. Thermometret hænger frit imod Nord.

Fig. 6. The first published Christiania weather table, from *Den norske Rigstidende*,
24 January 1818.

Fig. 7. The temperature
Esmark's evening
morning observation the
season (Dec-Feb).

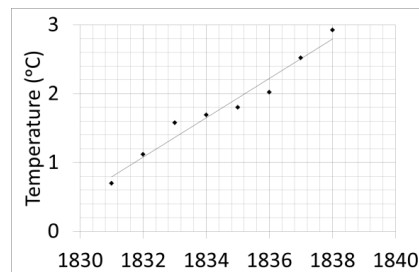

difference (ᵒC) between
observation and the
following day for the winter





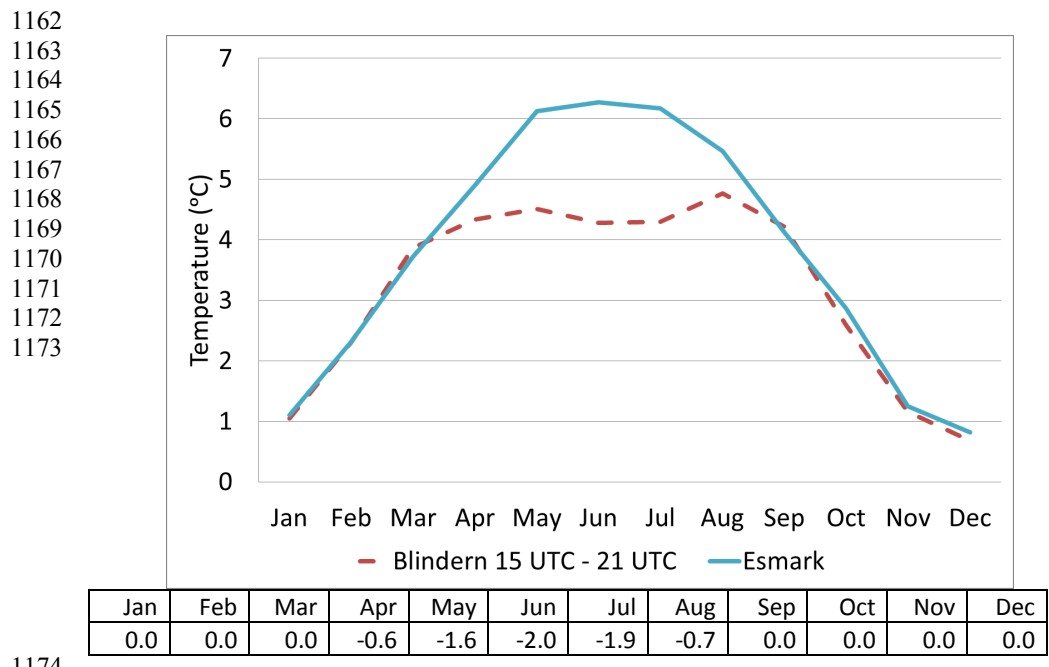

| Jan | Feb | Mar | Apr | May | Jun | Jul | Aug | Sep | Oct | Nov | Dec |
|------|------|------|------|------|------|------|------|------|------|------|------|
| 0.0 | 0.0 | 0.0 | -0.6 | -1.6 | -2.0 | -1.9 | -0.7 | 0.0 | 0.0 | 0.0 | 0.0 |

Fig. 8 Temperature differences (ºC) between the observations at Blindern at 15 UTC and at 21 UTC for the
period 1993.01-2015.09. Also the difference between the midday and evening observations of Esmark is shown
for the period 1816.01-1838.12. (The adjustments of the evening observations, Table 5, are added to the data for
the period 1816.01-1821.12 before the calculation of the differences. In the table below the figure are shown the
adjustments of Esmark's midday observation.






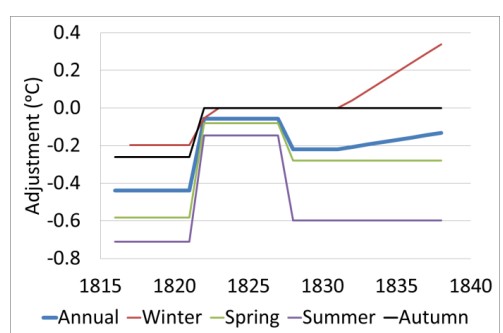

Fig. 9. Adjustments added to Esmark's series for each season during his period of
observation, 1816-1838.






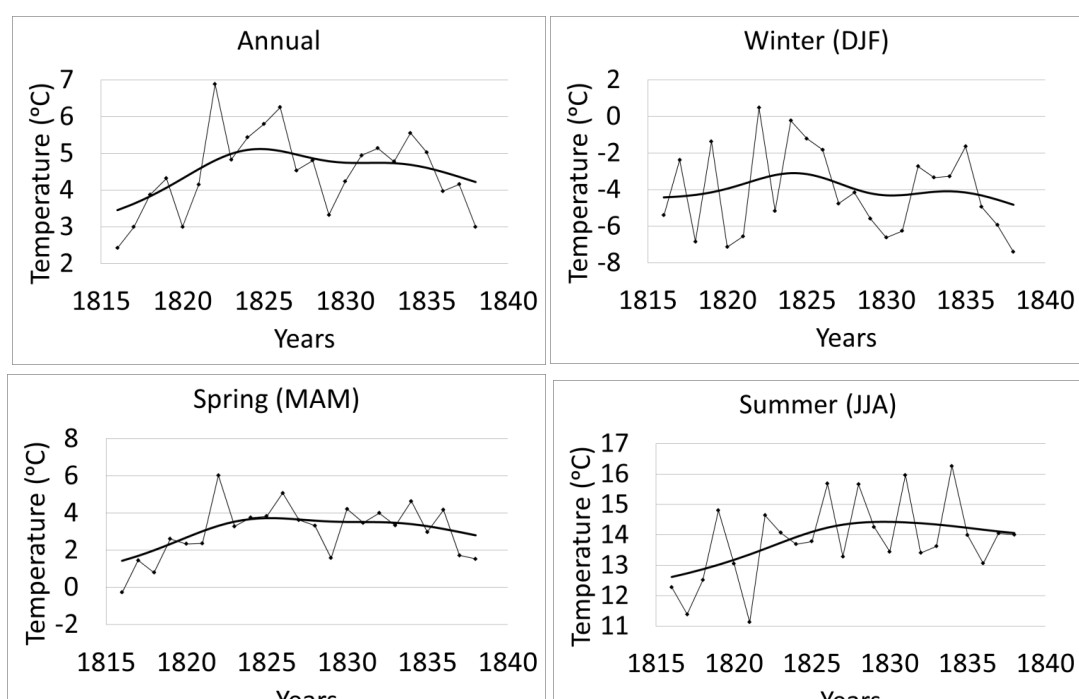

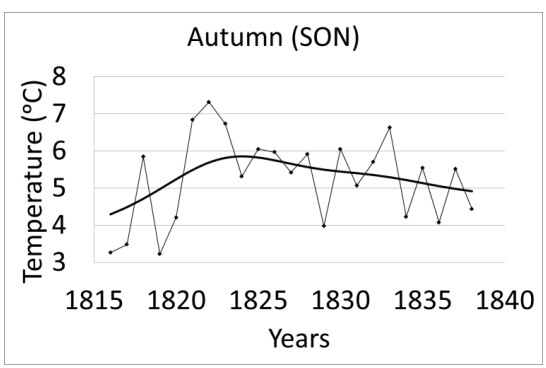

Fig. 10. Annual and seasonal means of Esmark's temperature series (symbols), and
Gaussian filter (curves) with standard deviation 3 in the Gaussian distribution (e.g. Nordli
et al., 2015), corresponsing roughly to a 10 year regtangular filter.





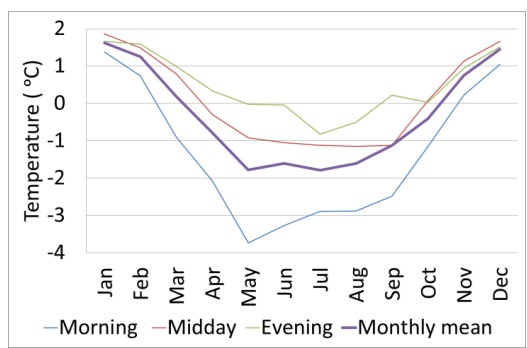

Fig. 11. Difference between Esmark's observations at Øvre Vollgate and Hansteen's
observations at Pilestredet (Esmark minus Hansteen) during the period 1822.11-1827.02 at
08, 15 and 21 UTC. The monthly means are calculated by Føyn's formula (see Appendix B).






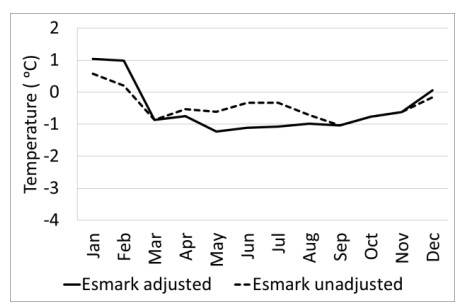

Fig. 12. Differences in mean monthly temperature between Esmark's observations at Øvre
Vollgate and those at the Astronomical Observatory (Esmark minus Observatory) during the
period 1837.04-1838.12. Esmark's observations are presented both unadjusted and adjusted.
For the observatory the temperatures are unadjusted.






