# Peer review of "Jens Esmark's Christiania (Oslo) meteorological observations 1816-1838: The first long term continuous temperature record from the Norwegian capital homogenized and analysed"

_Climate of the Past, 2016_

## Referee Comment (RC1) · Anonymous Referee #1 · 12 Jul 2016

The manuscript entitled "Jens Esmark's Christiania (Oslo) meteorological observations 1816-1838: The first long term continuous temperature record from the Norwegian capital homogenized and analysed" by Hestmark and Nordli describes the recovery of a new instrumental meteorological series for the early instrumental period, covering a particularly interesting time span for climate research (corresponding to the late Dalton minimum).

The authors put a considerable amount of effort in recovering metadata, which are

crucial for such old observations, and used them to aid the homogenisation of the temperature data. This is surely a good practice, however in many cases they had to turn to speculation to explain the causes of the inhomogeneities. In particular, they applied a correction for a possible overheating in the afternoon with very weak arguments to support it. I also have some reserve on the homogeneity testing, I think that reference series can and should be used. The other corrections applied should be improved, too.

In general I found the manuscript well organised, however it would benefit from an editing of the English language by a native speaker, because the structure of the sentences is often a bit weird and/or some punctuation is missing. The conclusions are poorly written and incomplete.

I recommend publication in Climate of the Past with minor revisions, in particular the homogenisation part needs in my opinion a bit of additional work.

See the attached pdf for additional comments.

Please also note the supplement to this comment:
http://www.clim-past-discuss.net/cp-2016-60/cp-2016-60-RC1-supplement.pdf

**Supplement:**

GENERAL COMMENTS

It is a pity that the analysis focuses on temperature only, especially considering that a lot of metadata are also provided for the other variables and that pressure data were corrected by Esmark himself (therefore should be of good quality). I suppose that the reason is that only temperature data have been digitised?
Pressure observations would be particularly useful to the research community, for instance for reanalysis projects (e.g., Compo et al., 2011). If they have been digitised, I would like to see in the manuscript at least some information on the pressure series concerning data quality. The series of Stockholm from the Bolin Centre could be used as reference, or one could also use pressure reconstructions (e.g., Luterbacher et al. (2002); Küttel et al. (2010)).

Reference series are not used for the homogeneity testing. The authors justify this choice with the unavailability of contemporary temperature observations near Oslo (Lines 355-356). However, I expect temperature series such as Bergen, Stockholm, Uppsala, and Copenhagen (all available in public datasets such as GHCN-M: https://www.ncdc.noaa.gov/ghcnm/) to be correlated enough to be used as reference series, at least at annual resolution. The absolute tests carried out by the authors give valuable information but could potentially overlook inhomogeneities due for example to changes in the instrumentation, therefore I think that the choice of not using reference series should be reconsidered and an additional relative homogeneity test should be applied after monthly means are calculated (Sect. 4.5).

SPECIFIC COMMENTS

1.INTRODUCTION

Line 67: Formally Austria-Hungary was only created in 1867. I suggest to use "in todays Slovakia".

Line 91: Replace "reanalyse" with "analyse".

Line 92: Add "temperature" before "data" (unless other variables are analysed in the revised version).

2.METADATA

Line 102: Remove "a" before "garden".

Lines 199-201: Please rephrase.

Line 214: What does the "average?" in parenthesis mean? You don't know if the published data are averages or you don't know if the corrections are applied to averages only? Either way, can this not be verified from the weather diaries?

Line 233: I think the translation for "hevertbarometer" is siphon barometer. Please verify.

Line 247: I think you mean "can" instead of "might".

Line 253: What is the unit of the pressure observations?

Line 256: drizzle.

Line 257: Does "Flokker" have no meaning?

Line 259: cloudy.

Line 273: "and" should be "a".

3.METHODS

Line 345: What is the significance level that you adopted?

Line 349: Replace "calculated" with "converted"

4.RESULTS

Line 355: "For much of Esmark's period of observation there was no other nearby station in operation so internal testing was the only possibility" I disagree on this. Monthly mean temperature anomalies at stations 400 km apart are usually still strongly correlated ($r>0.8$) (e.g., Auer et al., 2007). It should be possible to use data from Sweden, Denmark for reference and integrate the internal testing results with a relative test.

Line 367: "But this break in homogeneity was much less than that of the morning observation." Less what?

Line 398: Replace "notes" with "values".

Lines 418-419: Correct title and first line of the paragraph.

Line 449: What about the other seasons? According to your formula, you adjust the minimum temperature of 28 February 1838 by 2°C, while the minimum temperature of the following day is not corrected at all! I think here a correction function should be estimated for each month of the year (with some smoothing to better represent the annual cycle if the correction parameters are too noisy).
You also ignore other significant breakpoints without explaining why (e.g., 1835 for III vs II in spring in Table 2).

Line 453: I am not convinced by the evidences for an overheating of the midday observations. You compare with a station with arguably a very different microclimate (different elevation, distant from the sea, etc.). I think that your conclusions should be more conservative, considering the limited information you have on the thermometer exposition and the surrounding environment. You could say that a correction might be necessary for some applications (e.g., analysis of extremes), but for the analysis carried out in this manuscript I don't really see the point of applying such a correction.

5.DISCUSSION

Line 544: "Also the midday observation is warmer by Hansteen than by Esmark. This is harder to understand." Isn't this because of the overheating correction that you applied to Esmark's data?

Line 567: Here also the differences with the Astronomical Observatory in summer (Fig. 12) are in large part created by the overheating correction.

Line 589: A Section 5.4 should start here.

Line 590: Change "was probably the greatest one" to "was the largest one in terms of sulphur mass ejected (Oppenheimer, 2003)".

Line 591: The role of Tambora on the climate anomalies in Europe and North America is still debated. Besides, the "paradigm" of the Year Without a Summer is related not only to temperature, but also (and probably more) to precipitation and cloud cover anomalies.

Lines 605-606: Isn't it somewhat surprising that in Bergen, only 200 km or so from Oslo, 1816 was one of warmest years? Is this consistent with the instrumental temperature series of Bergen? Can you comment on the uncertainty of the reconstructions for individual years?

Line 608-616: You cannot reach conclusions on "weather patterns, excessive rain, frost, snowfall" in the summer of 1816 just by analysing the seasonal mean temperature. You should rather answer the question: How consinstent are Esmark's observations with the results shown by Luterbacher and Pfister (actually, the temperature reconstruction that they use is from Casty et al. (2005))? It would be interesting if you could add the series of the nearest gridpoint of that reconstruction in Fig. 10 and comment on the differences.

6. CONCLUSIONS

This section is incomplete and too synthetic, it should be much improved.

TABLES AND FIGURES

Table 8: It would be practical to have an additional column with the reference for each reconstruction.

Figure 4: This figure is useless. Remove it.

REFERENCES

Auer, Ingeborg, et al. "HISTALP—historical instrumental climatological surface time series of the Greater Alpine Region." *International Journal of Climatology* 27.1 (2007): 17-46.

Casty, C., D. Handorf, and M. Sempf. "Combined winter climate regimes over the North Atlantic/European sector 1766–2000." *Geophysical research letters* 32.13 (2005).

Compo, Gilbert P., et al. "The twentieth century reanalysis project." *Quarterly Journal of the Royal Meteorological Society* 137.654 (2011): 1-28.

Luterbacher, Jürg, et al. "Reconstruction of sea level pressure fields over the Eastern North Atlantic and Europe back to 1500." *Climate Dynamics* 18.7 (2002): 545-561.

Küttel, Marcel, et al. "The importance of ship log data: reconstructing North Atlantic, European and Mediterranean sea level pressure fields back to 1750." *Climate Dynamics* 34.7-8 (2010): 1115-1128.

Oppenheimer, Clive. "Climatic, environmental and human consequences of the largest known historic eruption: Tambora volcano (Indonesia) 1815." *Progress in physical geography* 27.2 (2003): 230-259.

---

## Referee Comment (RC2) · R. Przybylak (Referee) · 2 Aug 2016

More and more old meteorological data is needed, at best from the last millennium, to reliably estimate range (including natural) of changes and variability of climate in this time, as well as causes of climate changes. As a result, the idea of data rescue activity has been enthusiastically taken up by many scientists. In recent years a significant growth in this kind of activity can be seen, which is also reflected in a rising number of publications. The majority of them can be assigned to the new discipline which was

born within climatology and was termed "historical climatology". The reviewed paper belongs to this category. In a very detailed way, the authors present new meteorological series of data from Oslo rediscovered by them for period 1816–1838. Source data, quality control and homogeneity of data series are perfectly presented (with a very detailed description of metadata) and the best climatological knowledge was used to obtain the most reliable temperature data. For this purpose, modern techniques of quality control and homogenization procedures and methods have been successfully used. All different kinds of metadata information have also been used to correct the data as best as possible. In the improvement of data quality, both historical (parallel) observations) and contemporary series of data from Oslo have been used. In the paper, the authors give an extensive and complete, as well as scientifically and methodically correct, analysis of data elaboration and climate characteristics in Oslo for the study period. The paper is clearly written, well structured and well documented. One small weakness may be the fact that the paper is quite long; fortunately, not all documentation is included in the main body of the paper. I can only suggest removing Figure 4, which in my opinion is not important; Figure 5 is enough to present Esmark's Christiania protocol. I must say that the amount and quality of supplementary material is appropriate and that all the documentation placed there is very helpful towards understanding the results presented in the main body of the paper.

---

## Referee Comment (RC3) · Anonymous Referee #3 · 5 Aug 2016

This paper presents a very interesting and valuable data set which extends our knowledge of early 19th century climate in north western Europe. It is an important paper which I recommend to be published following some additional calculations and revisions. While the authors have made extensive efforts to account for data quality and to homogenize the readings for long term climatic analysis in the face of sparse metadata, I am particularly uneasy about the lack of information concerning the observation times, and it is my opinion that further analyses may help reduce this uncertainty.

[Figure]

In particular, the authors could make use of frequency analysis as exemplified by the work of Bergström and Moberg (2002) and Slonosky (2014) to compare Esmark's daily morning, afternoon and evening observations to the nearly 25 years of modern hourly observations mentioned in Table 1 and possibly obtain an approximate idea of the times of observation. It may be necessary to sub-divide the historical record for suspected changes in observation time derived from the SNHT analysis and to consider the possibility of observation times, especially in the morning, changing with the season, if this is supported by other metadata (e.g the statement of observation times quoted on lines 188-189). If probable times of observation can be established, the entire analysis will stand on much firmer ground. As it stands, there are many adjustments made on a statistical basis which add to the uncertainty of the final values of the observations, particularly given the differences seen when compared with other nearby observations.

The accounting of the adjustments due to inhomogeneities detected by the SNHT and other intra-series comparisons is extremely thorough and to be commended, but as is presented leaves the reader confused. A plethora of monthly adjustments is proposed in Tables 2-5, but it is not clear which adjustments were finally applied to which observations, the sequence of the adjustments nor whether the adjustments were applied to the daily data or to the monthly means. If daily, there will be artificial jumps between the end of one month and the beginning of the next - see Vincent et al (2002). In general, more use might be made of the advantages gained by having daily, rather than monthly, observations to analyze; much work has been done in the field of historical climatology in the past decade and more on analyzing daily observations directly.

The fairly large differences shown between these data and other nearby stations, less than 1 km away, also give reasons for concern about the final quality of the data. Comparisons with other series, such as Uppsala -Bergström and Moberg (2002) - and Stockholm, -Moberg and Bergström (2002), although a considerable distance away, may still give valuable indicators as to the character of each month and help decide which series in the comparisons are the more reliable.
Finally, the online archive where the data will be placed should be mentioned. Please see attached file for complete references.

Please also note the supplement to this comment:
http://www.clim-past-discuss.net/cp-2016-60/cp-2016-60-RC3-supplement.pdf
[Figure]

**Supplement:**

Review of "Jens Esmark's Christiania (Oslo) meteorological observations 1816-1838: The first long term continuous temperature record from the Norwegian capital homogenized and analysed"
Geir Hestmark and Øyvind Nordli
General remarks:

This paper presents a very interesting and valuable data set which extends our knowledge of early 19[th] century climate in north western Europe. It is an important paper which I recommend to be published following some additional calculations and revisions. While the authors have made extensive efforts to account for data quality and to homogenize the readings for long term climatic analysis in the face of sparse metadata, I am particularly uneasy about the lack of information concerning the observation times, and it is my opinion that further analyses may help reduce this uncertainty.

In particular, the authors could make use of frequency analysis as exemplified by the work of Bergström and Moberg (2002) and Slonosky (2014) to compare Esmark's daily morning, afternoon and evening observations to the nearly 25 years of modern hourly observations mentioned in Table 1 and possibly obtain an approximate idea of the times of observation. It may be necessary to sub-divide the historical record for suspected changes in observation time derived from the SNHT analysis and to consider the possibility of observation times, especially in the morning, changing with the season, if this is supported by other metadata (e.g the statement of observation times quoted on lines 188-189). If probable times of observation can be established, the entire analysis will stand on much firmer ground. As it stands, there are many adjustments made on a statistical basis which add to the uncertainty of the final values of the observations, particularly given the differences seen when compared with other nearby observations.

The accounting of the adjustments due to inhomogeneities detected by the SNHT and other intra-series comparisons is extremely thorough and to be commended, but as is presented leaves the reader confused. A plethora of monthly adjustments is proposed in Tables 2-5, but it is not clear which adjustments were finally applied to which observations, the sequence of the adjustments nor whether the adjustments were applied to the daily data or to the monthly means. If daily, there will be artificial jumps between the end of one month and the beginning of the next - see Vincent et al (2002). In general, more use might be made of the advantages gained by having daily, rather than monthly, observations to analyze; much work had been done in the field of historical climatology in the past decade or two on analyzing daily observations directly.

The fairly large differences shown between these data and other nearby stations, less than 1 km away, also give reasons for concern about the final quality of the data. Comparisons with other series, such as Uppsala -Bergström and Moberg (2002) - and Stockholm –Moberg and Berström (1997), although a considerable distance away, may still give valuable indicators as to the character of each month and help decide which series in the comparisons are the more reliable.

Finally, all the data, including the raw data, should be placed in an online archive.

Specific remarks:
Introduction, line 45 and thereafter: "protocol" usually refers to a method; it would be less confusing if the authors could use a word like "logbook" or "weather registers" if they mean the actual physical records of Esmark's weather observations.

Introduction and Section 2: An interesting and important synopsis of the observer, his location and environment, and his instruments.

Line 350: The authors should take note of Gauvin's 2012 article on the Réaumur thermometer:

The authors should be aware from Gauvin's work that theoretical adjustment of 1.25 for Réaumur to Celsius may not be accurate. This could help explain some of the large differences seen when comparing Esmark's values to the nearby observations in section 5.

Section 4.1: The SHNT results seem somewhat ambiguous. Can the SNHT be run on all the 7665 days of observations, rather than dividing up into months and seasons? This might give a clearer indication of the actual break date. If this is too large a number for computational purposes, the series could be tested on running sub-portions (i.e first six months, move forward three months and test next six month period, and so on). Testing on other variables such as pressure might also give a potential indication of a change in the positioning of the instruments. It may help to further divide section 4.1 into subsections dealing with all of the adjustments to each of the three observation times separately and consecutively.

Line 380: A synopsis of the shifts and dates for each of the observation series would make these clearer. What were the final adjustments made to each series? A table summarizing the actual adjustments applied and the order in which they were applied would be helpful.

Line 381: The authors appear to be postulating a replacement of an hourly observation in the morning with a minimum thermometer. Hourly temperature observations and minimum temperature observations are not the same entity. If the authors think that a minimum thermometer was in use, a new series labelled "minimum temperature" should be analyzed. Rather than an inhomogeneity, this is a new variable.

Line 390-2/871: This reasoning needs to be better explained, especially given the actual observation times are unknown. What does the description in the title of Table 4 "minimum temperature at 0800 UTC" mean?

Line 405: More specifics are needed to explain this conclusion: 26% of interpreted "minimum" values being higher than the evening temperature is a high proportion. This unusual temperature trend is a situation which could occur with the passage of frontal systems overnight. What is the proportion of such unusual diurnal temperature trends in the modern record?

Table 4/ Line 872:  What is the authors' interpretation of the negative summer differences for 1816-1828 and 1822-1828, compared to the modern differences? How are these differences changed with the selection of different observation times in the modern period (e.g. 0700, 0600, and sunrise?).  Why is the period 1816-1821 corrected but not 1822-1828? Are these results from before or after the application of the adjustment of the 1821 inhomogeneity?

Line 442: How does this weakened diurnal temperature wave affect the reasoning section 4.1 concerning the minimum thermometer?

Page 14, lines 447-451, Figure 5: Adjusting from one postulated unknown time to a second unknown time is a procedure beset with uncertainty, particularly as the linear trend does not appear to apply as well in the middle of the period, 1833-1836, when the points would give a much less steep slope.  Have the authors explored regressions and residuals for other, finer time resolutions than the three-month period shown in Fig 5?  What is the value of the sum of squares error? If better estimates of actual times of observation can't be made, some portion of the data may just have to be classified as unusable.

Section 4.5 Again, how and why are these adjustment values derived?  This is not clear.

Line 457, Figure 8: These adjustments should be presented in a Table separate from the Figure.

Line 482: Section 4.6 should be in the discussion section, while section 5.1 would perhaps be better placed as a summary in section 4.  The comparisons with other observers and discussion of the thermometer error would be better placed in a data quality and comparison section, with the climatic discussion in a separate section.

Line 541: Again, if we don't know the observation times, it's impossible to attribute the difference between the observers to a specific cause such as instrument location.

Line 550: is 2100 UTC after sunset in summer?

Figure 12: This would seem to suggest that the unadjusted values for Esmark are closer to the Observatory than the adjusted values.

Check grammar: line 273; 416;  419; 432…  The grammar is in numerous locations (insufficient time to enumerate here) somewhat awkward, for example putting a place indicator before a time indicator, when often in English the time clause precedes the location clause.

References:

Bergström, Hans, and Anders Moberg. "Daily air temperature and pressure series for Uppsala (1722–1998)." Improved Understanding of Past Climatic Variability from Early Daily European Instrumental Sources. Springer Netherlands, 2002. 213-252

Gauvin, J. F. (2012). The instrument that never was: inventing, manufacturing, and branding Réaumur's thermometer during the enlightenment. Annals of Science, 69(4), 515-549.

Moberg, A., Bergström, H., Krigsman, J. R., & Svanered, O. (2002). Daily air temperature and pressure series for Stockholm (1756–1998). In Improved Understanding of Past Climatic Variability from Early Daily European Instrumental Sources (pp. 171-212). Springer Netherlands.

Slonosky, V. (2014). Historical climate observations in Canada: 18th and 19th century daily temperature from the St. Lawrence Valley, Quebec. Geoscience Data Journal, 1(2), 103-120.

Vincent, L. A., Zhang, X., Bonsal, B. R., & Hogg, W. D. (2002). Homogenization of daily temperatures over Canada. Journal of Climate, 15(11), 1322-1334.)

---

## Author Response (AR1)

**UiO ⫶ Centre for Ecological and Evolutionary Synthesis**

University of Oslo

The Editor Climate of the Past

Re: Cp-2016-60, 2016

Date: 5 October 2016, Oslo

**Dear Editor**

Thank you for the positive and constructive referee reports on our paper on Jens Esmark's early temperature observations from the Norwegian capital. We hereby submit a revised version of our paper where due consideration has been made to the comments of the referees. More generally we have altered the structure of the Results chapter to make it more logical and readable, and changed several figures for the same reason.

Our answers to particular points raised by the referees are attached below, also to issues rise in last correspondence with Editor. We thus hope that the paper is substantially improved and will now be suitable for publication in Climate of the Past.

Sincerely yours

Geir Hestmark
Øivind Nordli

[Figure]

CEES, Dept. of Biosciences,
University of Oslo,
P.O. Box 1066 Blindern,
NO-0316 Oslo,
Norway, Europe

cees-post@bio.uio.no  www.mn.uio.no/cees

**To editorial remarks 28. Sept. 1016**

*You have opted not to perform relative testing with Stockholm and Copenhagen, but you now show the Stockholm series. However, visually the correlation seems very good. Does this support your argument that changes in climate patterns could matter here? Please comment on the correlations.*

--The correlation for annual values between the homogenized Stockholm series and the homogenized Esmark's series is 0.74, whereas correlation in the modern station networks of Norway is greater than 0.95. Without our corrections of the Esmark's series the correlation to Stockholm would have been less. The test does not tell us which station (or maybe both) of them that might have a possible inhomogeneity. We agree that the chance for a large inhomogeneity of the Stockholm series is less than for the Esmark's series because the Stockholm series is already tested against the nearby Uppsala series. However, the arguments against relative testing are strong. 1) The distance is 350 km between the stations so there is a risk for different climate development during a shorter time interval. 2) Neighbouring stations are only in a narrow sector. Therefore the situation described in 1) could easily occur 3) It would be a pity for climate research of the past if we miss important climate variability in the homogenisation process. There are certainly pro and contras in this issue, but we find the risk too high for "smoothing" away real variability

*Section 4.4. using the term "shift" might be misleading here as it is a trend.*

--We used a shift test so the only outcome of the test is "shift".  See also new comment on Fig 6  below.

*Tables: add unit of the shift,  perhaps mark those that were corrected in the end.*

--Units are added.

*Fig. 6: Explain the line.*

--Additional text is added, see 4.4

*All figures: Remove bounding box.*

--Done, and all figures are transferred to pdf-format

*Conclusions: A quick reader might conclude that the corrections can be applied to the daily scale. Please clarify, as the reviewers also repeatedly were asking you to consider the daily time scale.*

--The decision of using monthly and seasonal data for the homogenization process rather than daily or sub daily values was taken before we started. It seemed too ambiguous homogenizing Esmark's observations on daily data mainly for two reasons, i.e. lacking reference stations and uncertain observation times. Also, our main interest is to study the long term variability in climate, and later, establish an Oslo series starting with Esmark's observations. See also our answer to R3. If we have understood R3 correctly he thinks that using daily or in particular sub daily data should help us to better assess the observation times. However, the frequency distribution for neighbouring hours does not change much from hour to hour, so in this respect daily data will be of little help. What have helped us, is to study the difference of the mean values between temperatures at the midday observations (where temperature does not vary much from hour to hour) and temperatures at the morning and evening (where temperature vary much from hour to hour). The stability of this difference is our key method of this work. May be one should have used a robust mean, for example the median, rather than the arithmetic mean, but for the Esmark's observations this would not have given other results according to the quality control.

**Ad ref. Przybylak:**

- The original fig. 4 has been deleted as superfluous, and figures after no. 3 renumbered in accordance with this.

**Ad anonymous ref. 1:**

- The many suggestions for improvements of language and clarifications have all been adopted.

*Reference series are not used for the homogeneity testing. The authors justify this choice with the unavailability of contemporary temperature observations near Oslo (Lines 355-356). However, I expect temperature series such as Bergen, Stockholm, Uppsala, and Copenhagen (all available in public datasets such as GHCN-M: https://www.ncdc.noaa.gov/ghcnm/) to be correlated enough to be used as reference series, at least at annual resolution. The absolute tests carried out by the authors give valuable information but could potentially overlook inhomogeneities due for example to changes in the instrumentation, therefore I think that the choice of not using reference series should be reconsidered and an additional relative homogeneity test should be applied after monthly means are calculated (Sect. 4.5).*

We think it is important not to use reference stations too far away, in particular when they do not represent the whole circle around the station under testing. In this way wrong conclusions might be drawn as spatial temperature differences could be interpreted as inhomogeneities. For the Esmark series there exist contemporary stations mainly to the east (Stockholm/Uppsala) and south (Copenhagen). From Bergen original data are lacking, and daily data exist only in these intervals: 1818.01-1818.05, 1823.08-1824.07, 1824.08-1824.09, mostly printed in newspapers, and mean values, of questionable quality. We will not use Bergen as reference for the homogeneity testing as it could "contaminate" the results rather than improving them.

In the discussion part we have already compared Esmark's observations with other series from Oslo. Now this section is extended by binging in the Stockholm/Uppsala and the Copenhagen series. However, in the result part we exclude relative testing over those long distances (350 km and 450 km for Stockholm and Copenhagen respectively), also because (as mentioned above) the narrow sector available.

Taking into account some further work (see text) and also your consideration we have dropped the correction for overheating of the midday observation. We have also changed formula for calculating monthly mean temperature. Now, we use Mohn's formula, which is more robust when there is not complete knowledge of the observation times.

When correcting for the inhomogeneity in the evening observation, 1816-1821, a trivial sign error entered the original table submitted. This has been corrected.

*Line 355: "For much of Esmark's period of observation there was no other nearby station in operation so internal testing was the only possibility" I disagree on this. Monthly mean temperature anomalies at stations 400 km apart are usually still strongly correlated (r>0.8) (e.g., Auer et al., 2007). It should be possible to use data from Sweden, Denmark for reference and integrate the internal testing results with a relative test.*

Same point as above. If Esmark's series was given the same pattern of variability as the stations far away, important spatial climate variability could be hidden. However, we now open for using the available stations Stockholm/Uppsala and Copenhagen for evaluation of the final result, but detecting inhomogeneities by internal testing is our focus. It should be kept in mind that we do not know the exact observation times for the whole period of Esmark's observations. Here internal testing is the only possible way for detecting changes over time.

*Line 367: "But this break in homogeneity was much less than that of the morning observation." Less what?*

A reference to Table 2 is added.

*Line 449: What about the other seasons? According to your formula, you adjust the minimum temperature of 28 February 1838 by 2°C, while the minimum temperature of the following day is not corrected at all! I think here a correction function should be estimated for each month of the year (with some smoothing to better represent the annual cycle if the correction parameters are too noisy).*

We do not disagree, but prefer to keep this correction as simple as possible. The aim for the article is to homogenise a series of monthly mean temperatures only. Also, the correction is applied to minimum temperature only, so the amount of the correction on the monthly means will be much less than 2°C, cf. Fig. 7.

*You also ignore other significant breakpoints without explaining why (e.g., 1835 for III vs II in spring in Table 2).*

New text has been added.

*Line 453: I am not convinced by the evidences for an overheating of the midday observations. You compare with a station with arguably a very different microclimate (different elevation, distant from the sea, etc.). I think that your conclusions should be more conservative, considering the limited information you have on the thermometer exposition and the surrounding environment. You could say that a correction might be necessary for some applications (e.g., analysis of extremes), but for the analysis carried out in this manuscript I don't really see the point of applying such a correction.*

There are certainly good arguments for correcting the midday observations, but as you say we have limited knowledge of the microclimate at the station area. We have now made a new comparison with the station Oslo II, where thermometers were well protected by the Astronomical Institute building, and this supports no correction, Fig. 9.

*Line 544: "Also the midday observation is warmer by Hansteen than by Esmark. This is harder to understand." Isn't this because of the overheating correction that you applied to Esmark's data?*

No, the comparison was done before Esmark's data were corrected (However, now they are not corrected)

*Line 567: Here also the differences with the Astronomical Observatory in summer (Fig. 12) are in large part created by the overheating correction.*

Yes, an important reason for dropping the correction of the midday observation, see above.

*Line 591: The role of Tambora on the climate anomalies in Europe and North America is still debated. Besides, the "paradigm" of the Year Without a Summer is related not only to temperature, but also (and probably more) to precipitation and cloud cover anomalies.*

We now note these points in the Discussion

*Lines 605-606: Isn't it somewhat surprising that in Bergen, only 200 km or so from Oslo, 1816*

*was one of warmest years? Is this consistent with the instrumental temperature series of Bergen? Can you comment on the uncertainty of the reconstructions for individual years?*

Bergen is situated in quite another climate region than Oslo – oceanic vs. semi-continental.. The distance between the cities is about 300 km. See new comment in the text and reference to literature.

*Line 608-616: You cannot reach conclusions on "weather patterns, excessive rain, frost, snowfall" in the summer of 1816 just by analysing the seasonal mean temperature.*

Our conclusion concerns temperature only. However, it is important not to forget that Europe is larger than southern and western Europe. In the media we often see the summer of 1816 reckoned as the coldest one in our newest history. The present article contributes to a more nuanced view.

*You should rather answer the question: How consistent are Esmark's observations with the results shown by Luterbacher and Pfister (actually, the temperature reconstruction that they use is from Casty et al. (2005))? It would be interesting if you could add the series of the nearest gridpoint of that reconstruction in Fig. 10 and comment on the differences.*

Good point. We have done this. See new figure and text.

Re CONCLUSION: *This section is incomplete and too synthetic, it should be much improved.*

See new text

*Table 8: It would be practical to have an additional column with the reference for each reconstruction.*
Done

Relevant new references have been included.

**Ad anonymous ref. 2.**

*While the authors have made extensive efforts to account for data quality and to homogenize the readings for long term climatic analysis in the face of sparse metadata, I am particularly uneasy about the lack of information concerning the observation times, and it is my opinion that further analyses may help reduce this uncertainty. In particular, the authors could make use of frequency analysis as exemplified by the work of Bergström and Moberg (2002) and Slonosky (2014) to compare Esmark's daily morning, afternoon and evening observations to the nearly 25*

*years of modern hourly observations mentioned in Table 1 and possibly obtain an approximate idea of the times of observation.  It may be necessary to sub-divide the historical record for suspected changes in observation time derived from the SNHT analysis and to consider the possibility of observation times, especially in the morning, changing with the season, if this is supported by other metadata (e.g the  statement of observation times quoted on lines 188-189). If probable times of observation can be established, the entire analysis will stand on much firmer ground.  As it stands, there are many adjustments made on a statistical basis which add to the uncertainty of the final values of the observations, particularly given the differences seen when compared with other nearby observations.*

We think that the difference between a relatively stable midday temperature and the temperature at times of the day when temperature are changing most rapidly is the most efficient tool for detecting changes in observation times. On the other hand the shape of the frequency distribution does not change much from one hour to the next, so it will not help us much. The main problems of Bergstrøm and Moberg (2002) and for Slonosky (2014) are not observation times, but the environment of the thermometers. We agree that for those purposes (and in particular for the data quality control) frequency distribution analysis is very well suited. See also our comments to reviewer No. 1.

*The accounting of the adjustments due to inhomogeneities detected by the SNHT and other intra-series comparisons is extremely thorough and to be commended, but as is presented leaves the reader confused.  A plethora of monthly adjustments is proposed in Tables 2-5, but it is not clear which adjustments were finally applied to which observations, the sequence of the adjustments nor whether the adjustments were applied to the daily data or to the monthly means.  If daily, there will be artificial jumps between the end of one month and the beginning of the next - see Vincent et al (2002). In general, more use might be made of the advantages gained by having daily, rather than monthly, observations to analyze; much work had been done in the field of historical climatology in the past decade or two on analyzing daily observations directly.*

The Results chapter has been restructured to improve readability. We think that before starting the corrections of the observations it is important to have detected the inhomogeneities. This is what we have done, cf. Table 2 and Table 3. We agree that to interpret the tables might be challenging, but we want to give the reader the opportunity to be able to better judge our work. You claim that "it is not clear which adjustments were finally applied to which observations". We think we have now made this easier to follow, as we have first a detection part (4.1) and then the corrections are discussed for each of the three shifts (4.2, 4.3 and 4.4). We also made it clear that the homogeneity testing was done with monthly and seasonal values, so we have no aim of adjusting daily values. From this article the outcome will not be a series of homogenized daily values. However, we agree that it is important to specify what data and how the results will be stored in the database at the Norwegian Meteorological Institute. See new text.

*The fairly large differences shown between these data and other nearby stations, less than 1 km away, also give reasons for concern about the final quality of the data. Comparisons with other series, such as Uppsala -Bergström and Moberg (2002) - and Stockholm –Moberg and Berström (1997), although a considerable distance away, may still give valuable indicators as to the character of each month and help decide which series in the comparisons are the more reliable.*

Yes, we have now proceeded further with this comparison, see new text.

*Finally, all the data, including the raw data, should be placed in an online archive.*

Yes, they will. This is easy in Norway, where the entire network of stations is freely available for everybody.

*Introduction, line 45 and thereafter: "protocol" usually refers to a method; it would be less confusing if the authors could use a word like "logbook" or "weather registers" if they mean the actual physical records of Esmark's weather observations.*

As Webster's Dictionary defines a protocol as 'an original draft, minute or record' and as neither editor nor any of the other referees have had objections to our use of 'protocol' here, we think the term is appropriate as used.

*Line 350: The authors should take note of Gauvin's 2012 article on the Réaumur thermometer: The authors should be aware from Gauvin's work that theoretical adjustment of 1.25 for Réaumur to Celsius may not be accurate. This could help explain some of the large differences seen when comparing Esmark's values to the nearby observations in section 5.*

The problems related to the 'Reaumur scale' vs. other temperature scales were already thoroughly discussed in Middleton's book *A History of the Thermometer and Its Uses in Meteorology* (1966), where the conclusion is that by the 1780s a 'Reaumur scale' and 'Reaumur thermometer' had stabilized with thermometer manufacturers that was in fact rather different from the several scales and designs proposed by Reaumur, who died in 1757. For instance alcoholic solutions had been substituted by mercury, and the zero point defined by the melting rather than the freezing of ice etc. etc. Gauvin's paper mainly concerns the confusion surrounding this scale before this period, and is not particularly relevant (and largely recapitulates Middleton's work). The R-scales used by Esmark were certainly of the late 18[th] century kind, where notably the work of Swiss savant Jean-Andre De Luc had cleared up most of the confusion.

It is today a standard convention to convert R-values to C-values by the formula R x 1.25 = C, and although some R-thermometers may not exactly exhibit this calibration with the C, such nonlinear deviations would be much too small to explain the large value differences in Sect. 5.

*Section 4.1: The SHNT results seem somewhat ambiguous. Can the SNHT be run on all the 7665 days of observations, rather than dividing up into months and seasons? This might give a clearer indication of the actual break date. If this is too large a number for computational purposes, the series could be tested on running sub-portions (i.e first six months, move forward three months and test next six month period, and so on). Testing on other variables such as pressure might also give a potential indication of a change in the positioning of the instruments. It may help to further divide section 4.1 into subsections dealing with all of the adjustments to each of the three observation times separately and consecutively.*

Barometric data have yet to be digitized, and as stated in the metadata we have no reason to believe that barometers and thermometers were situated in exactly the same place.

*Line 380: A synopsis of the shifts and dates for each of the observation series would make these clearer. What were the final adjustments made to each series? A table summarizing the actual adjustments applied and the order in which they were applied would be helpful.*

See new text.

*Line 381: The authors appear to be postulating a replacement of an hourly observation in the morning with a minimum thermometer. Hourly temperature observations and minimum temperature observations are not the same entity. If the authors think that a minimum thermometer was in use, a new series labelled "minimum temperature" should be analyzed. Rather than an inhomogeneity, this is a new variable.*

We think our arguments for the minimum thermometer are very strong. We put the minimum temperature to the 'morning' observation only because listed by Esmark as such. Of course information will be to find in the metadata file of the MET-Norway.

*Line 390-2/871: This reasoning needs to be better explained, especially given the actual observation times are unknown. What does the description in the title of Table 4 "minimum temperature at 0800 UTC" mean?*

We have amended the text to provide a better explanation.

*Line 405: More specifics are needed to explain this conclusion: 26% of interpreted "minimum" values being higher than the evening temperature is a high proportion. This unusual temperature trend is a situation which could occur with the passage of frontal systems overnight. What is the proportion of such unusual diurnal temperature trends in the modern record?*

In a modern record the minimum temperature comes from the same sensor as the ordinary temperature so this percentage will be 0%. On manual stations it is difficult to say because the meteorological institutes correct there minimum thermometer with a so called "formal test". In practice this means that the minimum temperature will be corrected. In the case of Esmark one should expect more violation of this formal test as he seems not to have used a screen for the thermometers. For modern manual stations both thermometers are located in a common Stevenson screen.

*Table 4/ Line 872: What is the authors' interpretation of the negative summer differences for 1816-1828 and 1822-1828, compared to the modern differences? How are these differences changed with the selection of different observation times in the modern period (e.g. 0700, 0600, and sunrise?). Why is the period 1816-1821 corrected but not 1822-1828? Are these results from before or after the application of the adjustment of the 1821 inhomogeneity?*

Our first hypothesis was that changed observation time could explain the inhomogeneities, but this is hardly probable, see discussion. The answer to the second question is yes. See new text.

*Line 442: How does this weakened diurnal temperature wave affect the reasoning section 4.1 concerning the minimum thermometer?*

The effect is zero as the problem with the minimum thermometer only concerned the winter. See new text.

*Page 14, lines 447-451, Figure 5: Adjusting from one postulated unknown time to a second unknown time is a procedure beset with uncertainty, particularly as the linear trend does not appear to apply as well in the middle of the period, 1833-1836, when the points would give a much less steep slope. Have the authors explored regressions and residuals for other, finer time resolutions than the three-month period shown in Fig 5? What is the value of the sum of squares error? If better estimates of actual times of observation can't be made, some portion of the data may just have to be classified as unusable.*

The same procedure is used as for the other inhomogeneities, detection and then correction. For correction linear regression analysis is used with regression lines shown as equation (2). The standard error of the estimates is 0.15°C.

*Section 4.5 Again, how and why are these adjustment values derived? This is not clear.*

See the last answer and text.

*Line 457, Figure 8: These adjustments should be presented in a Table separate from the Figure.*

We think tables and regression equation is sufficient in addition to Fig. 7 that gives an overview.

*Line 482: Section 4.6 should be in the discussion section, while section 5.1 would perhaps be better placed as a summary in section 4. The comparisons with other observers and discussion of the thermometer error would be better placed in a data quality and comparison section, with the climatic discussion in a separate section.*

As these errors are not used in the analysis, they have been moved to Appendix B.

*Line 541: Again, if we don't know the observation times, it's impossible to attribute the difference between the observers to a specific cause such as instrument location.*

For Hansteen we know the observation times. (for Esmark we only know the exact observation time in 1833).

*Line 550: is 2100 UTC after sunset in summer?*

Yes

*Figure 12: This would seem to suggest that the unadjusted values for Esmark are closer to the Observatory than the adjusted values.*

Yes, this is one of the reasons for not having applied correction of Esmark's midday observation in this new version.

[revised manuscript text omitted]

Fig. 5

Moved (insertion) [22]

[Figure]

Fig. 6.

Field Code Changed

[Figure]

Fig. 7

Moved (insertion) [38]

Field Code Changed

**Annual**

Field Code Changed

**Spring (MAM)**

Field Code Changed

**Summer (JJA)**

Field Code Changed

**Autumn (SON)**

Field Code Changed

**Winter (DJF)**

Fig 8

Field Code Changed

[Figure]

Fig. 9

Moved (insertion) [39]

Field Code Changed

[Figure]

Fig. 10

Moved (insertion) [40]

Field Code Changed

Fig. 11.

Moved (insertion) [41]

Field Code Changed

Fig. 12

Moved (insertion) [42]

Field Code Changed

[Figure]

Moved (insertion) [43]

Fig. 13

---

## Author Response (AR2)

**UiO : Centre for Ecological and Evolutionary Synthesis**
University of Oslo

The Editor, Climate of the Past

Date: 17 October 2016

Dear Editor,

Thank you for your comment 14 October on our paper on Jens Esmark's Christiania-series. Re your wish to include Hertzberg's data from Ullensvang by the Hardangerfjord:

It would really have been a great advantage if we had a reliable series overlapping Esmark's series in the opposite direction to Stockholm. The problem with Hertzberg's observations is that the original data are lost. It is however known that Hertzberg observed various times during the day, from three to six times. For each day he calculated a "daily mean value". However, these values are not real daily means. It seems that he considered day temperature more important than night temperature, and it is not known how he performed his calculations.

With varying observation time during the day it is very likely that Hertzberg's "daily means" are inhomogenous. Also, Hertzberg was often on voyages so that there are gaps in his series. These were filled by Birkeland by the help of other stations, it is not clear which. It is understandable that MET Norway has given the digitalization of Hertzberg's "daily means" low priority so they are not digitized.

It would be easy for us to plot the annual means calculated by Birkeland in Fig. 12, but we are reluctant to do so because wee are afraid that it could mislead readers rather than add value to our article.

Birkeland was a pioneer for his time, so all honor to him. However, it is easy to forget how difficult a task he had with many of the early series. This is true for Ullensvang before 1865, Trondheim before 1856 and Bergen before 1860.

Sincerely yours

G. Hestmark & Ø. Nordli

[Figure]

CEES, Dept. of Biosciences,
University of Oslo,
P.O. Box 1066 Blindern,
NO-0316 Oslo,
Norway, Europe

cees-post@bio.uio.no  www.mn.uio.no/cees

[revised manuscript text omitted]

Fig. 1

Moved (insertion) [17]

Fig. 2

[Figure]

Fig. 3

[Figure]

Fig. 4

Meteorologiſke Jagttagelſer i Chriſtiania 1818, anſtillede af Prof. Esmark.

| Januar. | Barometret. | Thermom. | Veirliget. |
|---|---|---|---|
| 1 | 28 T. 3 L. | — $11\frac{1}{5}°$ | Taage og tykt Veir |
| 2 | 28 6¼ | — $10\frac{1}{4}$ | Skyet. |
| 3 | 28 6¾ | — $8\frac{2}{3}$ | Tykt Veir. |
| 4 | 28 5 | — $11\frac{1}{6}$ | Lidt Snee. |
| 5 | 28 1⅔ | — $9\frac{1}{3}$ | Lidt Snee. |
| 6 | 27 11⅔ | — $4\frac{1}{6}$ | Tykt og lidt Snee. |
| 7 | 27 6⅙ | ✠ ¾ | Tykt Veir. |
| 8 | 27 5⅙ | ✠ ¾ | Stærk Taage. |
| 9 | 27 10⅓ | — $4\frac{1}{4}$ | Taage. |
| 10 | 27 5¾ | ✠ 1¼ | Bl. af S., Nordlys |
| 11 | 27 6¼ | ✠ 1½ | Klart Veir. |
| 12 | 27 6¼ | ✠ ¼ | Sn. og Regn SV |
| 13 | 27 5⅙ | 0 | Sn. og Regn SV |
| 14 | 27 6⅓ | ✠ ½ | Klart. |
| 15 | 26 10⅓ | ✠ ½ | Snee og Bl. af S. |

Anmærkninger: Observationerne ere anſtilles de 34 Rhinlandſke Fod over Havet, og ere Middeltallet af Observationer, anſtillede Morgen, Middag og Aften. Barometer-Høiderne ere corrigerede ſaaledes, ſom de ſkulle være, derſom Barometret havde været udſat for 0° Temperatur. Thermometret hænger frit imod Nord.

Fig. 5

Field Code Changed

[Figure]

Fig. 6.

Moved (insertion) [37]

Field Code Changed

[Figure]

Fig. 7

Moved (insertion) [38]

Field Code Changed

Field Code Changed

Field Code Changed

Field Code Changed

Autumn (SON)

Field Code Changed

Winter (DJF)

Fig 8

Field Code Changed

[Figure]

Fig. 9

Moved (insertion) [39]

Field Code Changed

[Figure]

Fig. 10

Moved (insertion) [40]

Field Code Changed

Fig. 11.

Moved (insertion) [41]

Field Code Changed

Fig. 12

Moved (insertion) [42]

Field Code Changed

[Figure]

Moved (insertion) [43]

Fig. 13